# Understanding the Gaps in Satisficing Bandits

Chloé Rouyer [1]   Ronald Ortner [2]   Peter Auer [2]

## Abstract

We study a variant of the stochastic multi-armed bandit problem in which the learner aims to identify and play an arbitrary arm whose expected reward exceeds a known satisficing threshold $S$, rather than optimizing against the best arm. Prior work has shown that when such a satisficing arm exists, time-independent bounds on the satisficing regret are achievable, but these guarantees deteriorate when an arm lies close to the threshold. We focus on instances in which the excess gap $\Delta_*$ (gap between the best arm and the threshold) is small relative to the suboptimality gaps $\Delta_i$, a regime that exposes this limitation. To capture this challenge, we propose a new algorithm, `uncertain-UCB`, which achieves *satisficing* pseudo-regret of $O\left(\sum_{i:\,\Delta_i>\Delta_*} \frac{\ln(K/\Delta_*)}{\Delta_i}\right)$, while recovering standard pseudo-regret bounds when no arm exceeds the threshold. Further, we establish a near-matching lower bound in the small excess-gap regime, showing that any algorithm incurs at least $\Omega\left(\sum_{i:\,\Delta_i>\Delta_*} \frac{\ln\left(\frac{\Delta}{(K-1)\Delta_*}\right)}{\Delta_i}\right)$ satisficing pseudo-regret on some problem instances.

## 1. Introduction

The stochastic multi-armed bandit problem is a canonical model for studying the trade-off between exploration and exploitation in sequential decision-making (Thompson, 1933; Auer, 2002; Bubeck & Cesa-Bianchi, 2012; Slivkins, 2019; Lattimore & Svepesvári, 2020), and it forms a foundational building block of reinforcement learning. In the standard stochastic bandit setting, a learner is faced with the task of repeatedly selecting an action, called an arm, from a set of options. After selecting an arm, the learner observes only the reward of the chosen arm, and receives no information about the rewards of the others. The goal of the learner is to maximize their cumulative reward, where the rewards are stochastic, meaning that the rewards associated with each arm are sampled from fixed distributions that are unknown to the learner. Since the reward-generating process is fixed but unknown, performance is measured in terms of *regret*, defined as the difference between the learner's cumulative reward and that of the single arm with the highest expected reward, played at all rounds.

Accordingly, in the standard formulation, the learner aims to match as closely as possible the expected performance of the best arm. In many practical settings involving repeated decisions, however, optimality may be unnecessary or overly costly, and achieving a sufficiently good performance is often the primary objective. This perspective can be formalized by introducing a satisficing threshold and redefining the learner's objective to identify and repeatedly select any arm whose expected reward exceeds this threshold. This framework naturally arises in settings where meeting a prescribed performance level is more important than optimality, such as allocating tasks under a fixed budget, managing resource consumption subject to operational constraints, or aiming to meet a minimum performance standard rather than maximize outcomes. While satisficing objectives have been studied primarily in the bandit setting, they are particularly appealing in reinforcement learning, where the size and complexity of the decision space often render strict optimality impractical. Formally, as performance measure one usually considers the *satisficing regret* to which the learner contributes each time an arm with expected reward below the threshold $S$ is played.

Early investigations of satisficing in multi-armed bandits (Kohno & Takahashi, 2017; Tamatsukuri & Takahashi, 2019) show that when the satisficing threshold, referred to as an aspiration level, is chosen between the expected rewards of the best and second-best arm, the learner can achieve regret that does not grow with time, reflecting the fact that satisficing and optimal actions coincide in this regime. This assumption does not hold in general, since the learner has no control over either the threshold or the expected rewards of the arms. There may be several arms whose expected reward lies above the threshold, or none at all. Beyond the standard

[1]Institut für Mathematik, Universität Potsdam, Germany [2]Department Mathematik und Informationstechnologie, Montanuniversität Leoben, Austria. Correspondence to: Chloé Rouyer <chloerouyer.ml@gmail.com>.

*Proceedings of the 43rd International Conference on Machine Learning*, Seoul, South Korea. PMLR 306, 2026. Copyright 2026 by the author(s).

suboptimality gap $\Delta$, this motivates the introduction of two additional gap quantities that characterize the difficulty of satisficing bandit problems. Specifically, the *excess gap* $\Delta_*$ denotes the difference between the expected reward of the best arm and the threshold, while the *insufficiency gap* $\widetilde{\Delta}$ measures the difference between the threshold and the expected reward of an arm below the threshold. Understanding how these different gap parameters fundamentally limit and shape achievable satisficing guarantees is the central focus of this work.

In recent years, the literature on satisficing bandits has expanded, and Hüyük & Tekin (2021); Michel et al. (2023); Feng et al. (2025) introduce algorithms whose regret guarantees depend inversely on the excess gap $\Delta_*$ and the insufficiency gap $\widetilde{\Delta}$. When the insufficiency gap $\widetilde{\Delta}$ and the excess gap $\Delta_*$ are of different orders of magnitude, we can construct problem instances where the satisficing regret guarantees of each of these become vacuous, in the sense that they would scale with the extremely large inverse excess gap $\Delta_*^{-1}$ even when the suboptimality gap $\Delta$ is large. This behavior is particularly noticeable in the results of Michel et al. (2023); Feng et al. (2025), which are also built to be robust in the non-realizable regime (i.e., when all arms have expected rewards below the threshold). Accordingly, the problem instances that cause high regret are not the ones where the threshold is far above the expected reward of all the arms, but the instances where the threshold is right below the expected reward of the best arm, meaning that the excess gap $\Delta_*$ is small.

As the learner does not know the bandit problem ahead of time, it is not possible to guarantee a lower bound on $\Delta_*$. This can be problematic in practice: consider an easy bandit instance where the suboptimality gap $\Delta$ between the optimal and the sub-optimal arm is very large. Many classical standard bandit algorithms would achieve pseudo-regret guarantees of order $O(\frac{\ln T}{\Delta})$. However, if we provide the learner with the extra information that the expected reward of one of the two arms exceeds a given threshold, methods like the ones proposed by Michel et al. (2023); Feng et al. (2025) would have pseudo-regret guarantees that scale with $\Delta_*^{-1}$, which could be significantly larger than $\frac{\ln T}{\Delta}$ when $\Delta_*$ is small.

In this work, we investigate whether a dependency on $\Delta_*^{-1}$ or $\widetilde{\Delta}^{-1}$ is unavoidable in the satisficing regret and aim to gain a better understanding of the true dependencies of the satisficing pseudo-regret on the suboptimality gaps, the insufficiency gaps, and the excess gap.

Our contributions are summarized as follows:

- First, we propose a generalization of the lower bounds proposed by Michel et al. (2023) and Feng et al. (2025) to the $K$-armed bandit problem, and show that the

satisficing regret scales with $\Omega\left(\sum_{i:\widetilde{\Delta}_i>0}\frac{\widetilde{\Delta}_i}{\Delta_i^2}\right)$, which to the best of our knowledge is the first result formally proved for satisficing $K$-armed bandits for arbitrary $K \geq 2$. This result is formally stated in Theorem 3.3.

- One main contribution of this work is to show that no algorithm can match our general lower bound (Theorem 3.3) across all problem instances simultaneously, and that a dependency on either $\Omega\left(\frac{\ln\Delta_*^{-1}}{\Delta}\right)$ or $\Omega\left(\frac{\ln\widetilde{\Delta}^{-1}}{\Delta}\right)$ is unavoidable. The formal statement of this result can be found in Theorem 3.4.

- We propose a novel algorithm, uncertain-UCB, and show that it achieves near-optimal guarantees across the realizable and the non-realizable regimes simultaneously. In the realizable regime, it is the first algorithm whose pseudo-regret guarantees neither scale with $\Delta_*^{-1}$ nor with $\widetilde{\Delta}^{-1}$. Rather the satisficing pseudo-regret is of order $O\left(\sum_{i:\tilde{\Delta}_i>0}\frac{\widetilde{\Delta}_i}{\Delta_i^2}\ln\left(\frac{K}{\Delta_*}\right)\right)$, which is optimal up to a constant factor and a dependency on $K$ in the logarithm. The algorithm is presented in Section 4 and the bound on the satisficing pseudo-regret is given in Theorem 4.1.

- For a closer investigation of the critical phase regarding the pseudo-regret guarantees given by Michel et al. (2023); Feng et al. (2025), we take a closer look at what we call the transitional phase. It refers to the intermediate phase between the realizable and the non-realizable regime when $0 < \Delta_* \leq \frac{1}{\sqrt{T}}$. In this regime, the excess gap $\Delta_*$ becomes so small that a dependency on $\ln T$ becomes preferable to a dependency on $\ln\frac{1}{\Delta_*}$. We show that in that regime, our algorithm simultaneously enjoys satisficing pseudo-regret guarantees of order $O\left(\sum_{i:\tilde{\Delta}_i>0}\left(\frac{\widetilde{\Delta}_i}{\Delta_i^2}\ln T\right)\right)$ and standard pseudo-regret guarantees of order $O\left(\sum_{i:\Delta_i>0}\frac{\ln T}{\Delta_i}\right)$. We also show that this standard pseudo-regret guarantee holds in the non-realizable regime. These results are presented in Theorem 4.2.

## 1.1. Related Literature

One of the earliest references to satisficing bandits is from Reverdy et al. (2017), who introduced it in the Bayesian setting. Generalizing satisficing bandits to the frequentist framework, Hüyük & Tekin (2021) proposed a reduction of the satisficing bandit setting to a more general framework of bandits with lexicographically ordered objectives. The algorithm they suggest achieves a time-independent satisficing pseudo-regret bound of order $O\left(\sum_{i:\widetilde{\Delta}_i>0}\frac{\ln(1/\widetilde{\Delta}_i)}{\widetilde{\Delta}_i}\right)$. The algorithm and its analysis are based on similar results obtained by Garivier et al. (2019) for the classic bandit setting when the learner knows the optimal expected reward

$\mu_*$, which can be seen as a special case of satisficing bandits where the threshold contains even more specific information. These bounds and their relation to our results will be discussed in detail in Section 3 below.

Michel et al. (2023) further generalized this result, proposing the first algorithm with "satisfy if you can, optimize if you must" type guarantees. Their algorithm enjoys time-independent satisficing regret guarantees of order $O\left(\sum_{i:\mu_i < S} \widetilde{\Delta}_i + \frac{1}{\Delta_i} + \frac{\widetilde{\Delta}_i}{\Delta_*^2}\right)$ when the problem is realizable, meaning that one of the arms has an expected reward above the satisficing threshold. Simultaneously, it ensures a rate-optimal logarithmic upper bound on the pseudo-regret in the non-realizable case, when the means of all the arms are below the threshold $S$. Recalling that the suboptimality gap satisfies $\Delta_i = \Delta_* + \widetilde{\Delta}_i$, the bound proposed by Michel et al. (2023) is optimal when $\Delta_* \approx \widetilde{\Delta}_i$ for all insufficient arm $i$, but it degrades when either the excess gap or the insufficiency gap is very small compared to the full suboptimality gap.

More recently, Feng et al. (2025) have suggested an approach for satisficing bandits that combines a standard bandit algorithm with a lower confidence bound check. The bounds on the satisficing regret that are shown for this method are of order $O\left(\frac{K}{\Delta_*}\text{polylog}(\frac{1}{\Delta_*})\right)$. Note that $\Delta_*$ is smaller than any of the suboptimality gaps $\Delta_i$ of arms $i$ below the threshold. This highlights that the dependency on $\widetilde{\Delta}_i^{-1}$ in the results of Michel et al. (2023) is not necessary, and suggests that the satisficing regret should indeed scale with the inverse of the excess gap $\Delta_*^{-1}$. This algorithm adapts to the non-realizable regime, and the authors have also extended their results to provide the first satisficing pseudo-regret results for Lipschitz and concave bandits.

We would also like to point out some recent work on generalizing satisficing results from the bandit to the MDP setting by Hajiabolhassan & Ortner (2026). For reinforcement learning in a communicating MDP with finite state and action spaces, the proposed SAT-RL algorithm achieves time-independent satisficing regret, provided a satisficing policy exists. In the non-realizable case, results analogous to those of Michel et al. (2023) are obtained.

Concerning lower bounds, Bubeck et al. (2013, Theorem 5) and Garivier et al. (2019, Theorem 7) consider a bandit setting with two arms where the mean of the best arm $\mu^*$ is known and show that an asymptotic dependency on $\frac{1}{\Delta}$ is unavoidable, even when the gaps are unknown. Michel et al. (2023) suggest that these lower bounds should generalize to the satisficing regime as $\Omega\left(\sum_{i:\widetilde{\Delta}_i > 0} \frac{1}{\widetilde{\Delta}_i}\right)$, whereas Feng et al. (2025) prove that the regret should scale with $\Omega\left(\frac{1}{\Delta_*}\right)$. In their construction, all these lower bounds assume that the gaps $\Delta_* = \widetilde{\Delta}_i$ are the same for all arms $i$ below the threshold. These bounds focus on the global effect of the gaps on the regret, and even when they scale with $\Delta_*$, they do not consider the impact of the excess gap being small compared to $\Delta$.

## 2. Problem Setting and Notation

We consider a bandit problem with $K$ arms with respective expected rewards $\mu_1, \ldots, \mu_K$ played on $T$ rounds, where the learner is given a satisficing threshold $S$. Importantly, in general the learner does not know whether the mean of any arm is above the threshold. We say that a bandit problem is *S-realizable* if there exists an arm $i \in [K]$ such that $\mu_i > S$, and we use the term *sufficient* to refer to the arms $i$ such that $\mu_i \geq S$ and *insufficient* to denote the arms $i$ with $\mu_i < S$.

Furthermore, we assume that the rewards are generated by sub-Gaussian distributions. We recall that for a sub-Gaussian distribution with mean $\mu$ and empirical estimate $\hat{\mu}_n$ computed from $n$ samples it holds for any $\varepsilon > 0$:

$$\mathbb{P}\left[\hat{\mu}_n > \mu + \varepsilon\right] \leq e^{-n\varepsilon^2/2},$$
$$\mathbb{P}\left[\hat{\mu}_n < \mu - \varepsilon\right] \leq e^{-n\varepsilon^2/2}. \tag{1}$$

In this paper, we will be using two different notations for the empirical averages: $\hat{\mu}_i(t)$ refers to the empirical mean of arm $i$ at round $t$, whereas $\hat{\mu}_{i,n_i}$ refers to the empirical mean of arm $i$ after it was sampled $n_i$ times.

We use $i^*$ to denote an arbitrary optimal arm, and $\mu_* = \mu_{i^*}$ for the respective maximal mean reward. Further, $\bar{\mu} := \min_{i:\mu_i > S} \mu_i$ is the sufficient arm with the smallest expected reward. We define the *suboptimality gaps* between the best arm and any arm $i$ as $\Delta_i = \mu_* - \mu_i \geq 0$, and the *excess gap* between the best arm and the threshold as $\Delta_* = \mu_* - S$. These quantities are represented in Figure 1. In the proofs, we also generalize the excess gap to the other sufficient arms by defining for every sufficient arm $j$, $\hat{\Delta}_j = \mu_j - S$.

In the classic multi-armed bandit problem, the performance of the learner on problem $\nu$ after $T$ rounds is evaluated in terms of the pseudo-regret, defined as

$$R_T(\nu) := \sum_{i:\Delta_i > 0} \Delta_i \, \mathbb{E}[n_i(T)], \tag{2}$$

where $n_i(T)$ denotes the number of times arm $i$ is played in $T$ rounds.

To measure the performance of a satisficing algorithm in the realizable regime, we have to change the comparison point: At each round, instead of suffering the distance between the chosen insufficient arm $i$ and the best arm, we suffer the gap between the expected reward of arm $i$ and the threshold $S$, defined as $\widetilde{\Delta}_i = S - \mu_i$. Accordingly, we measure

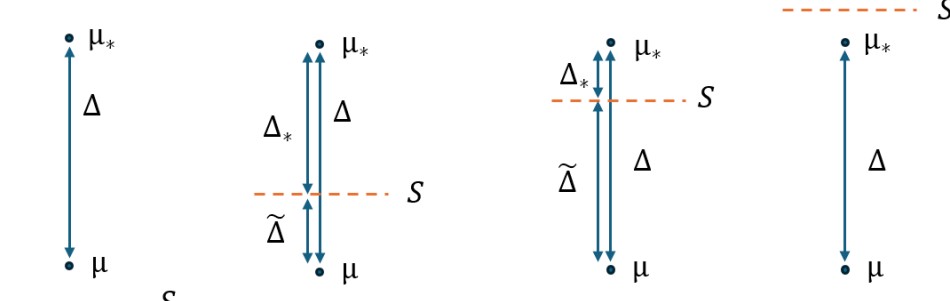

*Figure 1.* In this figure, we present four instances of 2-armed bandit problems, where $\mu_*$ and $\mu$ represent the expected rewards of the arms and $\mu^* > \mu$. We consider what happens when the threshold $S$ changes, while $\mu_*$ and $\mu$ remain the same:

1. On the left-most instance, both arms have an expected reward that is situated above the threshold $S$, so neither of the arms incurs satisfying pseudo-regret when played. Accordingly, the problem is trivial.

2. In the two central instances, the threshold is situated between the expected rewards of the arms. The problem is realizable. In one instance, the insufficiency gap $\widetilde{\Delta}$ is small compared to $\Delta$, whereas in the other the excess gap $\Delta_*$ becomes smaller.

3. On the right-most instance, both arms have an expected reward that is situated below the threshold $S$. The problem is non-realizable. The satisfying pseudo-regret is linear in this case, so we measure the classic pseudo-regret with respect to the best arm instead.

performance via

$$\mathcal{R}_T^S(\nu) = \sum_{i:\widetilde{\Delta}_i>0} \widetilde{\Delta}_i \mathbb{E}[n_i(T)]. \qquad (3)$$

In the literature, this measure is known as the *satisficing regret*, even though it should be more accurately called the *satisficing pseudo-regret*. When highlighting the threshold $S$ we also sometimes use the term $S$-*satisficing* instead of just *satisficing*. We often drop the dependency on $\nu$ in the notation for the satisficing regret when it is clear from the context.

## 3. Lower Bounds

In this section, we derive new lower bounds for the satisficing bandit problem in the realizable regime. Previous work, such as that of Michel et al. (2023) and Feng et al. (2025), derived bounds for the two-armed setting, providing lower bounds for the satisficing pseudo-regret that scale as $\Omega(\frac{1}{\Delta})$ and $\Omega(\frac{1}{\Delta_*})$, respectively.

The proof of Feng et al. (2025) assumes that the two arms are evenly spaced around the threshold, and is therefore not adapted to characterize the small excess gap regime, where $\Delta_*$ is small compared to $\Delta$. The first result we present is a generalization of these two previous results to $K$ arms, so that it applies to any realizable satisficing bandit problem. Following the proof structure presented in Garivier et al. (2019), this proof requires a mild constraint on the behavior of the algorithm.

**Definition 3.1.** An algorithm is considered *stable* if for any

bandit problem $\nu$ with two optimal arms $i, j \in [K]$ that have the same distribution $\nu_i = \nu_j$, it holds for all $t$:

$$\mathbb{E}[n_i(t)] = \mathbb{E}[n_j(t)].$$

This requirement is milder than requiring the algorithm to be stable under permutations of the arms.

**Definition 3.2.** An algorithm achieves time-independent satisficing pseudo-regret for a threshold $S$ if for any $S$-realizable bandit problem, the satisficing pseudo-regret satisfies

$$\mathcal{R}_T^S \leq C,$$

where $C$ can depend on all problem dependent quantities such as $\Delta$, $\widetilde{\Delta}$ and $\Delta_*$ but is independent of the time-horizon $T$.

**Theorem 3.3.** *For any stable algorithm that achieves time-independent satisficing pseudo-regret for a threshold $S$, there exists an $S$-realizable $K$-armed bandit problem $\nu$ for which this algorithm admits the following lower bound:*

$$\lim_{T\to\infty} \mathcal{R}_T^S(\nu) \geq \sum_{i:\widetilde{\Delta}_i>0} \frac{\widetilde{\Delta}_i}{\Delta_i^2}.$$

When there is a constant factor ratio between $\widetilde{\Delta}_i$ and $\Delta_*$, this lower bound shows that the algorithms proposed by Michel et al. (2023) are optimal up to multiplicative constants. The proof of Theorem 3.3 can be found in Appendix A.

We now derive an improved lower bound for the small excess gap regime $\Delta_i \gg \Delta_*$, which shows a logarithmic dependency on $1/\Delta_*$ is unavoidable in some regimes.

**Theorem 3.4** (Lower Bound for Small Excess Gaps). *Let $1 \geq \Delta > \Delta_* > 0$ such that the excess gap satisfies $\Delta_* = \gamma\Delta$ for some $0 < \gamma \leq \frac{1}{2\sqrt{K-1}}$, and consider a $K$-armed bandit problem $\nu$ with $\mu_* = \mu_1 := S + \Delta_* = S + \gamma\Delta$ and $\mu_j := \mu_1 - \Delta < S$ for $j = 2, \ldots, K$. Then, there is a $T > 0$ such that for any satisficing algorithm whose regret $\mathcal{R}_t^S(\nu)$ in $\nu$ after any $t \geq T$ steps is below*

$$\mathcal{R}_t^S(\nu) < \frac{(K-1)}{8} \cdot \frac{\ln\left(\frac{1}{(K-1)\gamma}\right)}{\Delta}$$
$$= \frac{(K-1)}{8} \cdot \frac{\ln\left(\frac{\Delta}{(K-1)\Delta_*}\right)}{\Delta},$$

*there exists a bandit problem $\nu'$ where $\mu_1' := S - \Delta_*$ and $\mu_i' := \mu_1' + \Delta$ for some $i > 1$, and $\mu_j' = \mu_j \ \forall j \notin \{1, i\}$. In this problem, arm $i$ is the unique sufficient arm, the new suboptimality gaps satisfy $\Delta_j' = \mu_i' - \mu_j' \in [\Delta, 2\Delta]$ for all $j \neq i$, the excess gap from problem $\nu$ is now the smallest insufficiency gap $\Delta_* = \widetilde{\Delta}_1' = \min_{j:\widetilde{\Delta}_j'>0}\left\{\widetilde{\Delta}_j'\right\}$ and*

$$\mathcal{R}_t^S(\nu') \geq \frac{(K-1)}{32} \cdot \frac{\ln\left(\frac{1}{(K-1)\gamma}\right)}{\Delta}$$
$$\geq \frac{1}{32} \sum_{j:\Delta_j'>0} \frac{\ln\left(\frac{\Delta_j'}{2(K-1)\widetilde{\Delta}_1'}\right)}{\Delta_j'}.$$

In the problem instance $\nu$, the statistical difficulty comes from the small excess gap $\Delta_*$, whereas in problem $\nu'$, the difficulty comes from the smallest insufficiency gaps $\widetilde{\Delta}_1' = \Delta_*$.

It is no accident that our lower bound is not instance-specific, which would hold for any specific parameters of the bandit problem. Instead, our lower bound only shows that for sub-optimality gap $\Delta$, a single algorithm cannot achieve $O\left(\frac{\ln(1/\Delta)}{\Delta}\right)$ regret for both the small excess gap regime and the small insufficiency gap regime. This is unavoidable, as our algorithm in Section 4 achieves $O\left(\frac{\ln(1/\Delta)}{\Delta}\right)$ regret for large excess gaps $\Delta_* \sim \Delta$, and Hüyük & Tekin (2021) achieve $O\left(\frac{\ln(1/\Delta)}{\Delta}\right)$ for large insufficiency gaps $\widetilde{\Delta} \sim \Delta$.

*Proof of Theorem 3.4.* Consider an instance $\nu$ of a satisficing $K$-armed bandit problem with threshold $S$, where the arms follow Gaussian distributions such that $\nu_i = \mathcal{N}(\mu_i, 1)$ for $i \in [K]$. Further, arm 1 is the unique optimal arm with mean $\mu_1 = S + \gamma\Delta$, while for all $j \neq 1$, we have $\mu_j = \mu_1 - \Delta < S$. Recall that we use $\widetilde{\Delta}_j = S - \mu_j = (1-\gamma)\Delta$

to denote the insufficiency gap of these arms. By construction, the suboptimal arms all have the same mean, so for all $j \neq 1$, $\Delta_j = \Delta$ and $\widetilde{\Delta}_j = \widetilde{\Delta}$.

We want to show that for $T$ sufficiently large, if an algorithm achieves low regret on the problem $\nu$, then we can construct an instance $\nu'$ for which the learner cannot achieve low regret simultaneously. Accordingly, we assume that we have

$$\mathcal{R}_t^S(\nu) = \sum_{j \neq 1} \widetilde{\Delta}_j \mathbb{E}_\nu[n_j(T)] \leq (K-1)\frac{\ln\alpha}{\Delta} \quad (4)$$

for all $t \geq T$ where we choose $T = \frac{\ln\alpha}{\Delta_*^2} = \frac{\ln\alpha}{\gamma^2\Delta^2}$ for a constant $\alpha$ to be defined later.

Equation (4) implies that there exists an arm $i \in \{2, \ldots, K\}$ such that $\mathbb{E}_\nu[n_i(T)] \leq \frac{\ln\alpha}{(1-\gamma)\Delta^2}$. This arm $i$ is chosen to construct another bandit instance $\nu'$, which is identical to the instance $\nu$, except for the distributions of arms 1 and $i$, which change their expected reward: we choose $\mu_i' = S + (1-\gamma)\Delta$ and $\mu_1' = \mu_i' - \Delta = S - \gamma\Delta$. In this new instance $\nu'$, arm $i$ is the unique sufficient and best arm, and arm 1 is insufficient. All the other arms remain unchanged, so $\mu_j' = \mu_j = \mu_i' - 2(1-\gamma)\Delta$ for all $j \notin \{1, i\}$. We note that in this problem $\nu'$, we now have:

$$\text{Excess gap: } \Delta_*' = (1-\gamma)\Delta,$$
$$\text{Suboptimality gaps: } \Delta_1' = \Delta$$
$$\Delta_j' = 2(1-\gamma)\Delta \quad \forall j \notin \{1, i\},$$
$$\text{Insufficiency gaps: } \tilde{\Delta}_1' = \gamma\Delta$$
$$\tilde{\Delta}_j' = (1-\gamma)\Delta \quad \forall j \notin \{1, i\},$$

Note that arm 1 does not have the same mean as the other insufficient arms, but under the assumption that $\gamma \leq 1/2$, we have for all $j \notin \{1, i\}$:

$$\Delta \leq 2(1-\gamma)\Delta = \Delta_j',$$

so $\Delta_j' \in [\Delta, 2\Delta]$ and $\Delta_1' = \Delta \in \left[1/2\Delta_j', \Delta_j'\right]$.

We show that a learner that achieves regret as in Equation (4) has to suffer large regret on the problem $\nu'$. To do so, we lower bound $\mathcal{R}_T^S(\nu')$.

Following a standard approach for lower bounds (Garivier et al., 2019, Equation 6), it holds that for any measurable random variable $Z \in [0, 1]$,

$$\sum_{j=1}^K \mathbb{E}_\nu[n_j(T)] \, \text{KL}(\nu_j, \nu_j')$$
$$= \mathbb{E}_\nu[n_1(T)] \, \text{KL}(\nu_1, \nu_1') + \mathbb{E}_\nu[n_i(T)] \, \text{KL}(\nu_i, \nu_i')$$
$$\geq \text{kl}\left(\mathbb{E}_\nu[Z], \mathbb{E}_{\nu'}[Z]\right), \quad (5)$$

where the first equality follows from $\forall j \notin \{1, i\}$, $\nu_j = \nu'_j$ and $\mathrm{KL}(\nu_j, \nu'_j) = 0$. We now derive an upper and a lower bound for Equation (5).

To upper bound the left hand side of Equation (5), we note that $\mathbb{E}_\nu[n_1(T)] \leq T = \frac{\ln \alpha}{\Delta_*^2}$ as no arm can be played more than the number of rounds, and $\mathbb{E}_\nu[n_i(T)] \leq \frac{\ln \alpha}{\Delta^2}$ by assumption that the learner achieves low regret on $\nu$ and that arm $i$ is sub-sampled. As $\mathrm{KL}(\nu_1, \nu'_1) = 2\Delta_*^2$ and $\mathrm{KL}(\nu_i, \nu'_i) \leq 2\Delta^2$ by Proposition A.1, we obtain

$$\mathbb{E}_\nu[n_1(T)] \, \mathrm{KL}(\nu_1, \nu'_1) + \mathbb{E}_\nu[n_i(T)] \, \mathrm{KL}(\nu_i, \nu'_i)$$
$$\leq 2\Delta_*^2 \frac{\ln \alpha}{\Delta_*^2} + 2\Delta^2 \frac{\ln \alpha}{\Delta^2} = 4 \ln \alpha. \tag{6}$$

Now choosing $Z = 1 - n_1(T)/T$ in (5) we obtain from (6) that:

$$\mathrm{kl}\left(\mathbb{E}_\nu[Z], \mathbb{E}_{\nu'}[Z]\right) \leq 4 \ln \alpha. \tag{7}$$

On the other hand, by Equation (11) of Garivier et al. (2019), it holds that $\mathrm{kl}(p, q) \geq (1 - p) \ln \left(\frac{1}{1-q}\right) - \ln 2$ for any $p, q \in (0, 1)$, which for our choice of $Z$ gives

$$\mathrm{kl}\left(\mathbb{E}_\nu[Z], \mathbb{E}_{\nu'}[Z]\right)$$
$$\geq \frac{\mathbb{E}_\nu[n_1(T)]}{T} \ln \left(\frac{T}{\mathbb{E}_{\nu'}[n_1(T)]}\right) - \ln 2. \tag{8}$$

Combining Equations (7) and (8) implies

$$\frac{\mathbb{E}_\nu[n_1(T)]}{T} \ln \left(\frac{T}{\mathbb{E}_{\nu'}[n_1(T)]}\right) - \ln 2 \leq 4 \ln \alpha. \tag{9}$$

In order to find a lower bound on $\frac{\mathbb{E}_\nu[(T)]}{T}$ first note that by our assumption defined in (4), we have

$$(K - 1)\frac{\ln \alpha}{\Delta} \geq R_T^S(\nu) = \widetilde{\Delta} \sum_{j \neq 1} \mathbb{E}_\nu[n_j(T)]$$
$$= (1 - \gamma)\Delta \sum_{j \neq 1} \mathbb{E}_\nu[n_j(T)].$$

Dividing by $(1 - \gamma)\Delta$ gives $\sum_{j \neq 1} \mathbb{E}_\nu[n_j(T)] \leq (K - 1)\frac{\ln \alpha}{(1-\gamma)\Delta^2}$, and we obtain by the choice of $T$ that

$$\mathbb{E}_\nu[n_1(T)] = T - \sum_{j \neq 1} \mathbb{E}_\nu[n_j(T)]$$
$$\geq \frac{\ln \alpha}{\gamma^2 \Delta^2} - (K - 1)\frac{\ln \alpha}{(1 - \gamma)\Delta^2}$$
$$\geq \frac{\ln \alpha}{\gamma^2 \Delta^2} - \frac{\ln \alpha}{2\gamma^2 \Delta^2}$$
$$= \frac{1}{2}T,$$

where the third step follows from our assumption that $\gamma \leq \frac{1}{2\sqrt{K-1}}$ and so $\frac{(K-1)}{(1-\gamma)} \leq \frac{1}{2\gamma^2}$. Together with (9) we obtain

$$\frac{1}{2} \ln \left(\frac{T}{\mathbb{E}_{\nu'}[n_1(T)]}\right) \leq 4 \ln \alpha + \ln 2 = \ln(2\alpha^4)$$

and consequently

$$\mathbb{E}_{\nu'}[n_1(T)] \geq \frac{T}{4\alpha^8} = \frac{\ln \alpha}{4\alpha^8 \gamma^2 \Delta^2},$$

which gives for the regret in instance $\nu'$

$$\mathcal{R}_T^S(\nu') \geq \widetilde{\Delta}'_1 \mathbb{E}_{\nu'}[n_1(T)] = \gamma\Delta \mathbb{E}_{\nu'}[n_1(T)] \geq \frac{\ln \alpha}{4\alpha^8 \gamma \Delta}, \tag{10}$$

noting that $\widetilde{\Delta}'_1 = \gamma\Delta$ is the suboptimality gap of arm 1 in $\nu'$.

Picking $\alpha = \left(\frac{1}{(K-1)\gamma}\right)^{1/8}$ satisfies

$$\frac{1}{\alpha^8 \gamma \Delta} = \frac{(K - 1)\gamma}{\gamma\Delta} \geq \sum_{j \neq i} \frac{1}{\Delta'_j},$$

where the last step follows from definition of the gaps in problem $\nu'$, which gives that $\Delta'_1 = \Delta$ and for all $j \notin \{1, i\}$, $\Delta'_j \in [\Delta, 2\Delta]$. To finish the Theorem, we lower bound $\alpha$ to express $\ln \alpha$ in terms of quantities in problem $\nu$ and $\nu'$ respectively. That is,

$$\alpha = \left(\frac{1}{(K - 1)\gamma}\right)^{1/8} = \left(\frac{\Delta}{(K - 1)\Delta_*}\right)^{1/8}$$
$$\geq \left(\frac{\Delta'_{\max}}{2(K - 1)\widetilde{\Delta}'_{\min}}\right)^{1/8},$$

where we recall that $\Delta'_j \in [\Delta, 2\Delta]$ for all $j$, so $\Delta'_{\max} = \max_j \Delta'_j \geq \Delta/2$ and $\widetilde{\Delta}'_{\min} = \widetilde{\Delta}'_1 \leq \widetilde{\Delta}'_j$ for $j \notin \{1, i\}$ is the smallest of the insufficiency gaps in problem $\nu'$. This finishes the proof as

$$\mathcal{R}_t^S(\nu') \geq \frac{(K - 1)}{32} \frac{\ln \left(\frac{1}{(K-1)\gamma}\right)}{\Delta}$$
$$\geq \frac{1}{32} \sum_{j: \Delta'_j > 0} \frac{\ln \left(\frac{\Delta'_{\max}}{2(K-1)\widetilde{\Delta}'_{\min}}\right)}{\Delta'_j}.$$

$\square$

## 4. An Algorithm for Satisficing Bandits

In this section, we introduce a novel `UCB1`-like algorithm and present upper bounds on its (satisficing) pseudo-regret in the realizable, non-realizable and transitional regimes.

**Algorithm 1** `uncertain-UCB`

---

**Input:** Number of arms $K$, threshold $S$, constants $C_1$ and $C_2$.

**Initialize:** Play each arm once, i.e., for time steps $t = 1, \ldots, K$ choose $i(t) = t$.

**for** $t = K + 1$ **to** $T$ **do**

   Compute:

$$n_0(t) = \sum_{\tau < t} \mathbb{I}\left[\hat{\mu}_{i(\tau)}(\tau) < S + \sqrt{C_2 \frac{\ln n_{i(\tau)}(\tau)}{n_{i(\tau)}(\tau)}}\right],$$

$$\forall i \in [K]: \ u_i(t) = \hat{\mu}_i(t) + \sqrt{C_1 \frac{\ln n_i(t) + \ln n_0(t)}{n_i(t)}}.$$

   Play $i(t) = \mathrm{argmax}_{i \in [K]} u_i(t)$.

   Observe reward of arm $i(t)$ and update the empirical estimate $\hat{\mu}_{i(t)}$.

**end for**

---

We define `uncertain-UCB` in Algorithm 1. While this algorithm follows the structure of the `UCB1` algorithm, the key difference between the two methods lies in the construction of the upper confidence bounds: Instead of using the current step count $t$ in the confidence intervals we use the number of *uncertain rounds*, defined as

$$n_0(t) = \sum_{\tau < t} \mathbb{I}\left[\hat{\mu}_{i(\tau)}(\tau) < S + \sqrt{C_2 \frac{\ln n_{i(\tau)}(\tau)}{n_{i(\tau)}(\tau)}}\right], \quad (11)$$

where $\hat{\mu}_i(\tau)$ is the empirical average reward of arm $i$ at round $\tau$, $i(\tau)$ is the arm played by the learner at round $\tau$, $n_i(\tau)$ is the number of times arm $i$ has been played up to round $\tau$ and $C_2$ is an input parameter of the algorithm, further specified in the theorems below.

This variable $n_0(t)$ presents the interesting characteristic of only increasing when the learner plays an arm whose empirical estimate is not sufficiently far above the threshold $S$. This means that as long as the learner cannot be reasonably certain that the arms that are played are satisficing, $n_0(t)$ keeps on increasing and remains close to $t$. During this phase, the algorithm behaves like the classic `UCB1`. Once the algorithm consistently plays arms that are sufficiently far above the threshold, the number of uncertain rounds $n_0(t)$ stops increasing and the upper confidence bounds of the arms stop growing. This approach slows and eventually stops the exploration of insufficient arms, which allows the algorithm to achieve time-independent satisficing regret.

The following theorem moreover shows that the upper bound on the satisficing regret in the realizable case is close to the lower bound of Theorem 3.4.

**Theorem 4.1.** *For any threshold $S$ and any $S$-realizable $K$-armed bandit problem,* `uncertain-UCB` *with $C_1 = 10$ and $C_2 = 12$ satisfies for all $T > 1$,*

$$\mathcal{R}_T^S \leq O\left(\sum_{i:\widetilde{\Delta}_i > 0} \left(\frac{\widetilde{\Delta}_i}{\Delta_i^2} \ln\left(\frac{K}{\Delta_*}\right) + \frac{\widetilde{\Delta}_i}{K} + \widetilde{\Delta}_i\right)\right).$$

The detailed proof of Theorem 4.1 is given in Appendix C. Here we present a proof sketch which highlights the main challenges of the proof.

*Proof Sketch.* To upper bound the satisficing pseudo-regret of `uncertain-UCB`, we need to bound the number of times each insufficient arm $i$ is played. Recalling that we defined

$$u_i(t) = \hat{\mu}_i(t) + \sqrt{C_1 \frac{\ln n_i(t) + \ln n_0(t)}{n_i(t)}},$$

apart from using a larger constant $C_1$ than for `UCB1`, the key difference between `uncertain-UCB` and `UCB1` is the use of $\ln n_i(t) + \ln n_0(t)$ instead of $\ln t$ in the numerator. We have to show that $n_0(t)$ is bounded. To do so, for all $j \in [K]$, we introduce $n_{0,j}(t)$ which is the contribution of arm $j$ to $n_0(t)$ and consider the event

$$\mathcal{E}_i(t) := \left\{n_{0,i}(t) \geq \frac{n_i(t)}{2}\right\},$$

that is, arm $i$ is played more often in uncertain rounds than in rounds that are not uncertain. Using a similar approach as in the standard analysis of the `UCB1` algorithm, we decompose the number of times an insufficient arm $i$ is played as:

$$\sum_{t=1}^{T} \mathbb{P}[i(t) = i]$$

$$\leq N_i + \sum_{t=1}^{T} \mathbb{P}[n_i(t) > N_i, i(t) = i, \ \mathcal{E}_i^c(t)]$$

$$+ \underbrace{\sum_{t=1}^{T} \mathbb{P}[i(t) = i, n_i(t) > N_i, u_{i^*}(t) \leq \mu_{i^*} - \Delta_i/2, \ \mathcal{E}_i(t)]}_{\text{Cases where the best arm } i^* \text{ underperforms}}$$

$$+ \underbrace{\sum_{t=1}^{T} \mathbb{P}[i(t) = i, n_i(t) > N_i, u_i(t) \geq \mu_i + \Delta_i/2, \ \mathcal{E}_i(t)]}_{\text{Cases where arm } i \text{ overperforms}}.$$

$$(12)$$

The first sum is easily bounded by a constant in Lemma C.2 as it turns out that an insufficient arm is mostly played in uncertain rounds.

For the analysis of the second sum, we need to deal with the dependence of $u_{i^*}(t)$ on $n_0(t)$, which in general depends on all arms. However, as we are bounding the probability of the best arm $i^*$ having a small upper confidence value $u_{i^*}(t)$, we only require a lower bound on $n_0(t)$. By picking $n_0(t) \geq n_{0,i}(t)$ and recalling that under $\mathcal{E}_i(t)$, $n_{0,i}(t) \geq n_i(t)/2$, we can control this term taking a union bound over the values $n_{i^*}(t)$ can have at rounds where arm $i$ is played, just as in the analysis of UCB1. This result is stated in Lemma C.3.

Finally, the key challenges of this proof come with bounding the last of the sums in Equation (12), which is detailed in Lemma C.4. This time, we need to upper bound $n_0(t)$, which means that we need to have upper bounds on $n_{0,i}(t)$ for all $i$ in $K$ simultaneously. To handle that, we introduce another event

$$M(t,n) := \left\{ \max_{i \in [K]} n_{0,i}(t) \leq n \right\},$$

for any parameter $n > 0$. This term allows us to upper bound $n_0(t)$, since $\max_{i \in [K]} n_{0,i}(t) \leq n$ obviously implies $n_0(t) \leq Kn$. For any insufficient arms $i$, we can reuse the same argument as in Lemma C.3 and show that $n_{0,i}(t)$ is proportional to $n_i(t)$. Then, we can argue that arm $i$ does not contribute much to $n_0(t)$ because it is not played often. Intuitively, we recall that to distinguish between two arms that have an expected gap of $\Delta$, approximately $\Delta^{-2}$ samples are needed. Accordingly, we expect arm $i$ to be played approximately $\Delta_i^{-2}$ times, which is its contribution to $n_0$.

For the sufficient arms however, we need a different approach. Intuitively, it would make sense to pick a value of $n$ that depends on the distance between the sufficient arm and the threshold $S$. As a sufficient arm $i$ has expected reward above threshold $S$, it would contribute to the uncertainty for approximately $\hat{\Delta}_i^{-2}$ rounds, where $\hat{\Delta}_i = \mu_i - S$ is the specific excess gap of sufficient arm $i$. The problem is that we have no control over how small $\hat{\Delta}_i$ might be, even compared to the global excess gap $\Delta_*$. Accordingly, the key trick of the analysis is to decompose the sufficient arms into two categories: the sufficient arms $i$ such that $\hat{\Delta}_i \geq \Delta_*/2$ and the others:

▷ Then for any sufficient arm $i$ with $\hat{\Delta}_i \geq \Delta_*/2$, we can use the just outlined argument that their contribution to $n_0(t)$ is bounded by a term of order $\hat{\Delta}_i^{-2} \leq 4\Delta_*^{-2}$.

▷ Otherwise, if arm $i$ is sufficient with $\hat{\Delta}_i < \Delta_*/2$, it means that $\mu_i$ is closer to $S$ than to $\mu_*$ so that $\Delta_i \geq \Delta_*/2$. In turn, we can argue that these arms are played at most $\Delta_i^{-2} \leq 4\Delta_*^{-2}$ times.

Overall, this shows that the total contribution of sufficient arms to $n_0(t)$ is of order $K\Delta_*^{-2}$ with high probability. Lemma C.6 bounds the third sum of Equation (12) when

this upper bound on $n_0(t)$ holds, while Lemmas C.7 and C.8 handle the cases where $n_0(t)$ is larger than $K\Delta_*^{-2}$.

Combining all these parts of the analysis, we deduce that

$$\sum_{t=1}^{T} \mathbb{P}\left[i(t) = i\right] \leq O\left(\frac{1}{\Delta_i^2} \ln\left(\frac{K}{\Delta_*}\right) + \frac{1}{K}\right),$$

and plugging this result into the definition of the satisficing pseudo-regret then finishes the proof. □

Now, in the two other regimes where the problem instance is not realizable, or the excess gap is small compared to the time horizon, uncertain-UCB can be shown to behave similarly to UCB1, which provides a smooth transition between the realizable and non-realizable cases.

**Theorem 4.2.** *For any non-realizable $K$-armed bandit problem* uncertain-UCB *with $C_1 \geq 4$ and $C_2 \geq 6$ satisfies for all $T > 1$,*

$$R_T \leq \sum_{i:\Delta_i > 0} \left(8C_1 \frac{\ln T}{\Delta_i} + 24\Delta_i\right).$$

*Further, in the transitional case, when there is an arm above the threshold $S$, but $T \leq \frac{C_1}{4\Delta_*^2}$, the following two bounds hold simultaneously:*

$$R_T \leq \sum_{i:\Delta_i > 0} 8C_1 \frac{\ln T}{\Delta_i} + \sum_{i:\Delta_i > \Delta_*} 24\Delta_i + \sum_{i:0 < \Delta_i \leq \Delta_*} 72\Delta_i$$

$$and \quad \mathcal{R}_T^S \leq \sum_{i:\widetilde{\Delta}_i > 0} \left(8C_1 \frac{\widetilde{\Delta}_i}{\Delta_i^2} \ln T + 24\widetilde{\Delta}_i\right).$$

In the transitional case, the excess gap tends to $0$ so that it becomes preferable to use $\ln T$ rather than $\ln\left(\frac{K}{\Delta_*}\right)$ in the upper bound. The proof of these results can be found in Appendix D.

## 5. Conclusion

We studied satisficing multi-armed bandits through the lens of gap-dependent complexity, showing that existing guarantees combine distinct sources of difficulty arising from suboptimality, insufficiency, and excess gaps. Our refined lower bounds reveal intrinsic tradeoffs that prevent uniform robustness across these regimes. Building on this analysis, we proposed an algorithm that achieves near-optimal guarantees in both realizable and non-realizable settings, while recovering standard bandit rates as the threshold approaches the optimal arm.

## Acknowledgements

The authors would like to thank the anonymous reviewers for their thoughtful comments which greatly helped

to improve the paper. This research was funded by the Austrian Science Fund (FWF) 10.55776/PAT6918624 and TAI 590-N and by the Deutsche Forschungsgemeinschaft (DFG)- Project-ID 318763901 - SFB1294, Project A03.

## Impact Statement

This paper presents work whose goal is to advance the field of theoretical Machine Learning. There are many potential societal consequences of Machine Learning in general, and while theoretical works such as ours could be used in a wide variety of applications, we don't anticipate that any negative impact would follow from this work.

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

# A. Proof of the Lower Bound in Theorem 3.3

In this section, we provide a proof for the lower bound presented in Theorem 3.3, following the analysis of the proof of Garivier et al. (2019, Theorem 7).

We use $\mathrm{KL}(P, Q)$ to denote the standard Kullback–Leibler divergence between distributions $P$ and $Q$ and $\mathrm{kl}(p, q)$ to denote the Kullback–Leibler divergence between two Bernoulli distributions with respective parameters $p$ and $q$.

The following result is standard and stated here for completeness.

**Proposition A.1.** *Consider $\nu_1 = \mathcal{N}(\mu_1, 1)$ and $\nu_2 = \mathcal{N}(\mu_2, 1)$ two uni-variate Gaussian distributions with unit variance, such that $|\mu_1 - \mu_2| = \Delta$. Then,*

$$\mathrm{KL}(\nu_1, \nu_2) = \frac{\Delta^2}{2}.$$

*Proof of Theorem 3.3.* Consider an instance $\nu$ of a satisficing $K$-armed bandit problem with threshold $S$ with a unique best arm $i^*$, where the arms follow Gaussian distributions such that $\nu_i = \mathcal{N}(\mu_i, 1)$ for $i \in [K]$, and the means fulfill $\mu_{i^*} = S + \Delta_*$ and $\mu_i = \mu_{i^*} - \Delta_i < S$ for all $i \neq i^*$.

As the learner achieves time-independent regret, it means that the number of times each insufficient arm is played in $\nu$ is constant. We want to lower bound the number of times each insufficient $i$ arm is played in $\nu$.

To do so, for each such arm $i$ we consider another bandit problem $\nu'$ defined such that $\forall j \neq i$, $\nu'_j = \nu_j$ and $\nu'_i = \nu_{i^*}$. This choice of $\nu'$ fulfills two important properties. First, the two problems differ only on arm $i$, meaning that

$$\sum_{j=1}^{K} \mathbb{E}_\nu[n_j(T)] \, \mathrm{KL}(\nu_j, \nu'_j) = \mathbb{E}_\nu[n_i(T)] \, \mathrm{KL}(\nu_i, \nu'_i) \geq \mathrm{kl}(\mathbb{E}_\nu[Z], \mathbb{E}_{\nu'}[Z]), \tag{13}$$

where the last step follows from Garivier et al. (2019, Equation 6) and holds for any bandit problems $\nu$, $\nu'$ and $\sigma(I_T)$-measurable random variables $Z$ with values in $[0, 1]$, where $I_T$ contains all the past information obtained by the algorithm up to round $T$. This holds in particular when picking $Z = n_i(T)/T$, as we will do in the following.

As $\nu_i = \mathcal{N}(\mu_i, 1)$ and $\nu'_i = \mathcal{N}(\mu_{i^*}, 1)$, we can ensure that $\mathrm{KL}(\nu_i, \nu'_i) = \frac{\Delta_i^2}{2}$. Combining these two statements with Equation (13), we obtain

$$\frac{\Delta_i^2}{2} \mathbb{E}_\nu[n_i(T)] = \mathbb{E}_\nu[n_i(T)] \, \mathrm{KL}(\nu_i, \nu'_i) \geq \mathrm{kl}\left(\frac{\mathbb{E}_\nu[n_i(T)]}{T}, \frac{\mathbb{E}_{\nu'}[n_i(T)]}{T}\right). \tag{14}$$

Then, in setting $\nu'$, we know that both arm $i$ and $i^*$ have the same distribution, meaning that Definition 3.1 ensures that $\mathbb{E}_{\nu'}[N_i(T)] = \mathbb{E}_{\nu'}[N_{i^*}(T)]$. As these arms are the two only sufficient arms and our algorithm achieves time-independent regret guarantees, all the other arms are played at most a constant number of times and we have

$$\lim_{T \to \infty} \frac{\mathbb{E}_{\nu'}[n_i(T)]}{T} = \frac{1}{2}. \tag{15}$$

Rearranging Equation (14) and taking the limit on $T$ gives:

$$\begin{aligned}
\lim_{T \to \infty} \mathbb{E}_\nu[n_i(T)] &\geq \frac{2}{\Delta_i^2} \lim_{T \to \infty} \mathrm{kl}\left(\frac{\mathbb{E}_\nu[n_i(T)]}{T}, \frac{\mathbb{E}_{\nu'}[n_i(T)]}{T}\right) \\
&\geq \frac{4}{\Delta_i^2} \lim_{T \to \infty} \left(\frac{\mathbb{E}_\nu[n_i(T)]}{T} - \frac{\mathbb{E}_{\nu'}[n_i(T)]}{T}\right)^2 \\
&\geq \frac{4}{\Delta_i^2} \left(\lim_{T \to \infty} \frac{\mathbb{E}_\nu[n_i(T)]}{T} - \lim_{T \to \infty} \frac{\mathbb{E}_{\nu'}[n_i(T)]}{T}\right)^2 \\
&\geq \frac{4}{\Delta_i^2} \left(0 - \frac{1}{2}\right)^2 \\
&= \frac{1}{\Delta_i^2},
\end{aligned}$$

where the first step reorganizes Equation (14), the second step applies Pinsker's inequality, which ensures that for any $p, q \in [0,1]$, $\mathrm{kl}(p, q) \geq 2(p - q)^2$. Then, the third step uses the properties of the limit: as both $\lim_{T \to \infty} \frac{\mathbb{E}_\nu[n_i(T)]}{T} = 0$ and $\lim_{T \to \infty} \frac{\mathbb{E}_{\nu'}[n_i(T)]}{T} = \frac{1}{2}$ are finite, and the square function is continuous, we can move the limits inside the square function.

We conclude that in problem $\nu$, insufficient arm $i$ is played in expectation $\mathbb{E}_\nu[n_i(T)] \geq \frac{1}{\Delta_i^2}$ times. This holds for any insufficient arm and the claimed lower bound on the regret follows. $\qquad \square$

## B. Analysis of the `uncertain-UCB` Algorithm

A key feature of the `uncertain-UCB` algorithm is the introduction of the number of uncertain rounds

$$n_0(t) := \sum_{\tau < t} \mathbb{I}\left[\hat{\mu}_{i(\tau)}(\tau) < S + \sqrt{C_2 \frac{\ln n_{i(\tau)}(\tau)}{n_{i(\tau)}(\tau)}}\right],$$

which quantifies how certain the learner is about having identified a sufficient arm, and which is used to adapt to the realizable and the non-realizable regimes. We also consider the number of uncertain rounds in which a particular arm $j$ has been played, that is,

$$n_{0,j}(t) := \sum_{\tau < t} \mathbb{I}\left[i(\tau) = j, \hat{\mu}_j(\tau) < S + \sqrt{C_2 \frac{\ln n_j(\tau)}{n_j(\tau)}}\right]. \tag{16}$$

To analyze the uncertainty of each arm individually, we generalize the notion of excess gap by defining for every sufficient arm $j$, $\hat{\Delta}_j = \mu_j - S$. Then, we note that for all $j$ and $t \geq 1$, $n_{0,j}(t)$ is always upper-bounded by the total number of times an arm is played, $0 \leq n_{0,j}(t) \leq n_j(t)$. We refine this bound depending on the relation between $\mu_j$ and $S$:

- If arm $j$ is insufficient, meaning that its expected reward is lower than the threshold $S$, we show that whenever arm $j$ is played, it will increase the uncertainty, meaning that with high probability, $n_j(t) - n_{0,j}(t)$ is proportional to $n_j(t)$. This result is proved in Lemma B.2.

- If arm $j$ is sufficient, we show that with high probability the number of uncertain rounds $n_{0,j}(t)$ remains upper bounded proportionally to $1/\hat{\Delta}_j$ even when $t$ tends to infinity. This result is proved in Lemma B.1.

- Furthermore, when the time horizon $T$ is small compared to $1/\Delta_*^2$ we show that we do not have enough samples to reliably distinguish between the expected reward of sufficient arm $j$ and the threshold $S$, so that for all $t \leq T$, $n_j(t) - n_{0,j}(t)$ is proportional to $n_j(t)$. This result is proved in Lemma B.3.

The following result is integral to the analysis of `uncertain-UCB` in the realizable case.

**Lemma B.1.** *When `uncertain-UCB` is run with $C_2 \geq 6$, the probability that sufficient arm $j$ contributes more than $n$ uncertain rounds is bounded as*

$$\mathbb{P}[n_{0,j}(t + 1) > n] \leq \frac{2 \exp\left(-(n - 1)\hat{\Delta}_j^2/2\right)}{\hat{\Delta}_j^2},$$

*where $\hat{\Delta}_j := \mu_j - S$.*

*Proof of Lemma B.1.* For any sufficient arm $j$, we recall that $\hat{\Delta}_j := \mu_j - S$ and have

$$\mathbb{P}\left[n_{0,j}(t+1) > n\right]$$

$$= \mathbb{P}\left[\exists s : i(s) = j, n_{0,j}(s) = n, \hat{\mu}_j(s) < S + \sqrt{C_2 \frac{\ln n_j(s)}{n_j(s)}}\right]$$

$$\leq \sum_{n_j \geq n} \mathbb{P}\left[\exists s : i(s) = j, n_j(s) = n_j, \hat{\mu}_j(s) < S + \sqrt{C_2 \frac{\ln n_j}{n_j}}\right]$$

$$\leq \sum_{n_j \geq n} e^{\frac{-n_j}{2}\left(\hat{\Delta}_j + \sqrt{C_2 \frac{\ln n_j}{n_j}}\right)^2}$$

$$\leq \sum_{n_j \geq n} e^{-\frac{n_j \hat{\Delta}_j^2}{2}}$$

$$\leq \frac{2 \exp\left(-(n-1)\hat{\Delta}_j^2/2\right)}{\hat{\Delta}_j^2}.$$

The first step uses the definition of $n_{0,j}$. As we don't know how many times $j$ is played, the second step takes a union bound over all possible values of $n_j(s)$, noting that $n_j(s) \geq n_{0,j}(s) = n$. In the third step, we use the fact that our rewards are sub-Gaussian. In the fourth step, we simplify using that $\exp(-x)$ is a decreasing function of $x$ and that $(a+b)^2 \geq a^2 + b^2 \geq a^2$ for $a, b \geq 0$. The last step bounds the sum by an integral, as for any $c > 0$ we have $\sum_{n_i \geq n} \exp(-cn_i) \leq \lim_{x \to \infty} \int_{n-1}^x \exp(-cn_i)dn_i = \frac{\exp(-c(n-1))}{c}$. $\qquad\square$

The following result is used in the analysis of `uncertain-UCB` in the realizable, non-realizable and transitional regimes. In particular, in the non-realizable regime, we note that all suboptimal arms are also insufficient, as all their expected rewards are below the satisficing threshold $S$. This result is the base of the proof that $n_0(t)$ scales with $t$ in that regime.

**Lemma B.2.** *When* `uncertain-UCB` *is run with $C_2 \geq 6$, the probability that after $t$ rounds insufficient arm $i$ has been played in more than $n$ not uncertain rounds is bounded as*

$$\mathbb{P}\left[n_i(t+1) - n_{0,i}(t+1) > n\right] \leq \frac{2}{n^{C_2/2 - 1}}.$$

*Proof of Lemma B.2.* For any insufficient arm $i$:

$$\mathbb{P}\left[n_i(t+1) - n_{0,i}(t+1) > n\right]$$

$$\leq \sum_{n_i \geq n} \mathbb{P}\left[\exists s : i(s) = i, n_i(s) = n_i, \hat{\mu}_i(s) \geq S + \sqrt{C_2 \frac{\ln n_i}{n_i}}\right]$$

$$\leq \sum_{n_i \geq n} \mathbb{P}\left[\exists s : i(s) = i, n_i(s) = n_i, \hat{\mu}_i(s) \geq \mu_i + \sqrt{C_2 \frac{\ln n_i}{n_i}}\right]$$

$$\leq \sum_{n_i \geq n} e^{\frac{-C_2 \ln n_i}{2}}$$

$$\leq \frac{2}{n^{C_2/2 - 1}}.$$

The first step counts the number of not uncertain rounds where arm $i$ is played, and we use a union bound over the possible values of $n_i(s)$. In the second step, we upper bound by a looser condition as $S \geq \mu_i$. This allows the use of the properties of sub-Gaussian distributions. To finish the proof, we use that $C_2/2 \geq 2$ so $\frac{1}{n_i^{C_2/2}}$ can be integrated, and we upper bound the sum by an integral $\sum_{n_i \geq n} \frac{1}{n_i^{C_2/2}} \leq \lim_{x \to \infty} \int_{n-1}^x \frac{1}{n_i^{C_2/2}} = \frac{1}{(C_2/2 - 1)(n-1)^{C_2/2 - 1}} \leq \frac{1}{(n-1)^{C_2/2 - 1}} \leq \frac{2}{n^{C_2/2 - 1}}.$ $\qquad\square$

Finally, we can extend this result to the case where the best arm is satisficing but $\Delta_*^{-2}$ is large compared to $T$. This allows to derive a smooth transition between the realizable and the non-realizable regimes.

**Lemma B.3.** *For any $n \geq 3$, when* `uncertain-UCB` *is run with $C_2 \geq 6$ and $T \leq \frac{C_2}{4\Delta_*^2}$, the probability that after $t \leq T$ rounds sufficient arm $i$ has been played in more than $n$ rounds that are not uncertain is bounded as*

$$\mathbb{P}\left[n_i(t+1) - n_{0,i}(t+1) > n\right] \leq \frac{2}{n^{C_2/2-1}}.$$

*Proof of Lemma B.3.* For any sufficient arm $i$

$$\begin{aligned}
&\mathbb{P}\left[n_i(t+1) - n_{0,i}(t+1) > n\right] \\
&\leq \sum_{n_i \geq n} \mathbb{P}\left[\exists s : i(s) = i, n_i(s) = n_i, \hat{\mu}_i(s) \geq S + \sqrt{C_2 \frac{\ln n_i}{n_i}}\right] \\
&\leq \sum_{n_i \geq n} \mathbb{P}\left[\exists s : i(s) = i, n_i(s) = n_i, \hat{\mu}_i(s) \geq \mu_i + \frac{1}{2}\sqrt{C_2 \frac{\ln n_i}{n_i}}\right] \quad (17) \\
&\leq \sum_{n_i \geq n} e^{\frac{-C_2 \ln n_i}{8}} \\
&\leq \frac{8}{n^{C_2/2-1}}.
\end{aligned}$$

This analysis follows the same steps as the proof of Lemma B.2 with the only difference appearing in Equation (17). We recall that for any sufficient arm $i$, $S = \mu_i - \hat{\Delta}_i$ with $\hat{\Delta}_i \geq 0$. Then Equation (17) holds if $\hat{\Delta}_i \leq \frac{1}{2}\sqrt{C_2 \frac{\ln n_i}{n_i}}$. By construction, $\hat{\Delta}_i \leq \Delta_* \leq \sqrt{\frac{C_2}{4T}}$ and $n_i \leq T$. As we sum over $n_i \geq n$ and $n \geq 3 > e$, we have $\frac{n_i}{\ln n_i} \leq n_i \leq T$, so that $\hat{\Delta}_i \leq \frac{1}{2}\sqrt{C_2 \frac{\ln n_i}{n_i}}$ indeed holds. The end of the proof follows by the same arguments as for Lemma B.2. $\square$

## C. Upper Bound in the Realizable Case

In this section we prove the upper bound on the satisficing regret for `uncertain-UCB` in the realizable setting. Let us first restate Theorem 4.1 with its exact constants.

**Theorem C.1.** *For any realizable $K$-armed bandit problem,* `uncertain-UCB` *with $C_1 = 10$ and $C_2 = 12$ satisfies for all $T > 1$,*

$$\mathcal{R}_T^S \leq \sum_{i:\widetilde{\Delta}_i > 0} \left(1294 \frac{\widetilde{\Delta}_i}{\Delta_i^2} \ln\left(\frac{640K}{\Delta_*^2}\right) + \frac{54336\widetilde{\Delta}_i}{K} + 16\widetilde{\Delta}_i\right).$$

To prove this result, we need to bound the number of times each insufficient arm $i$ is played, which we upper bound by a sum of different events. In particular, we define the event

$$\mathcal{E}_i(t) := \left\{n_i(t) - n_{0,i}(t) \leq \frac{n_i(t)}{2}\right\} = \left\{\frac{n_i(t)}{2} \leq n_{0,i}(t)\right\}, \quad (18)$$

and use $\mathcal{E}_i^c(t)$ to denote its complement. These two events are used to distinguish between the case where the estimates $n_{0,i}(t)$ and $n_i(t)$ behave as expected so that $n_{0,i}(t) < \frac{n_i(t)}{2}$, and the case where insufficient arm $i$ is overly contributing to the uncertainty, i.e., $n_{0,i}(t) \geq \frac{n_i(t)}{2}$. Using a standard decomposition with respect to $\mathcal{E}_i(t)$, the number of times insufficient

arm $i$ is played can be bounded as

$$\sum_{t=1}^{T} \mathbb{P}\left[i(t) = i\right] \leq N_i + \sum_{t=1}^{T} \mathbb{P}\left[n_i(t) > N_i, i(t) = i\right]$$

$$\leq N_i + \sum_{t=1}^{T} \mathbb{P}\left[n_i(t) > N_i, i(t) = i, \mathcal{E}_i(t)\right] + \sum_{t=1}^{T} \mathbb{P}\left[n_i(t) > N_i, i(t) = i, \mathcal{E}_i^c(t)\right]$$

$$\leq N_i + \sum_{t=1}^{T} \mathbb{P}\left[n_i(t) > N_i, i(t) = i, \mathcal{E}_i^c(t)\right]$$

$$+ \underbrace{\sum_{t=1}^{T} \mathbb{P}\left[i(t) = i, n_i(t) > N_i, u_{i^*}(t) \leq \mu_{i^*} - \Delta_i/2, \mathcal{E}_i(t)\right]}_{\text{Cases where the best arm } i^* \text{ underperforms}}$$

$$+ \underbrace{\sum_{t=1}^{T} \mathbb{P}\left[i(t) = i, n_i(t) > N_i, u_i(t) \geq \mu_i + \Delta_i/2, \mathcal{E}_i(t)\right]}_{\text{Cases where arm } i \text{ overperforms}}, \tag{19}$$

where each term can be bounded according to the following lemmas separately. It is important to note that this decomposition holds for any value of $N_i \geq 1$, but we can derive better bounds for the sums when $N_i$ is large.

The three sums of Equation (19) are bounded in the next three lemmas. The first lemma is also used in the analysis of `uncertain-UCB` in the non-realizable and transitional regimes.

**Lemma C.2.** *Consider any sub-Gaussian $K$-armed bandit problem. Using `uncertain-UCB` with $C_1 \geq 4$ and $C_2 \geq 6$, for any insufficient arm $i$ and any $N_i \geq 1$, it holds that*

$$\sum_{t=1}^{T} \mathbb{P}\left[i(t) = i, n_i(t) > N_i, \mathcal{E}_i^c(t)\right] \leq \frac{2}{(n_i/2)^{C_2/2-1}} \leq 16.$$

*Furthermore when $T \leq \frac{C_2}{4\Delta_*^2}$, for any arm $i$ (either insufficient or sufficient), we have*

$$\sum_{t=1}^{T} \mathbb{P}\left[i(t) = i, n_i(t) > N_i, \mathcal{E}_i^c(t)\right] \leq \frac{8}{(n_i/2)^{C_2/2-1}} \leq 64.$$

The next lemma bounds the probability that insufficient arm $i$ is played because optimal arm $\mu^*$ is underperforming and its upper confidence bound $u_{i^*}(t)$ is below the target value of $\mu_{i^*} - \Delta_i/2$.

**Lemma C.3.** *Consider any sub-Gaussian $K$-armed bandit problem. Using `uncertain-UCB` with $C_1 \geq 4$ and $C_2 \geq 6$, for any insufficient arm $i$ and any $N_i \geq 1$, we have*

$$\sum_{t=1}^{T} \mathbb{P}\left[i(t) = i, n_i(t) > N_i, u_{i^*} \leq \mu_{i^*} - \Delta_i/2, \mathcal{E}_i(t)\right] \leq \frac{21}{\Delta_i^2}.$$

**Lemma C.4.** *Consider any sub-Gaussian $K$-armed bandit problem and set $N_i = \frac{1280}{\Delta_i^2} \ln\left(\frac{640K}{\Delta_*^2}\right)$. Then using `uncertain-UCB` with $C_1 = 10$ and $C_2 = 12$ it holds for any insufficient arm $i$ that*

$$\sum_{t=1}^{T} \mathbb{P}\left[n_i(t) > N_i, i(t) = i, u_i(t) \geq \mu_i + \Delta_i/2, \mathcal{E}_i(t)\right] \leq \frac{64}{\Delta_i^2} + \frac{54336}{K}.$$

The proofs of these lemmas are given in the following section. In particular, the analysis of Lemma C.4 is detailed in Section C.2. With all these results available, we can move on to the proof of Theorem 4.1.

*Proof of Theorem 4.1.* We aim to derive a bound for the satisficing pseudo-regret defined as

$$\mathcal{R}_T^S = \sum_{i:\widetilde{\Delta}_i > 0} \widetilde{\Delta}_i \mathbb{E}[n_i(T)] = \sum_{i:\widetilde{\Delta}_i > 0} \widetilde{\Delta}_i \sum_{t=1}^T \mathbb{P}\left[i(t) = i\right].$$

For each insufficient arm $i$, we want to bound $\sum_{t=1}^T \mathbb{P}\left[i(t) = i\right]$. Following the decomposition in Equation (19), we have for arbitrary $N_i$ that

$$\sum_{t=1}^T \mathbb{P}\left[i(t) = i\right] \le N_i + \sum_{t=1}^T \mathbb{P}\left[n_i(t) > N_i, i(t) = i, \mathcal{E}_i^c(t)\right]$$

$$+ \sum_{t=1}^T \mathbb{P}\left[i(t) = i, n_i(t) > N_i, u_{i^*}(t) \le \mu_{i^*} - \Delta_i/2, \mathcal{E}_i(t)\right]$$

$$+ \sum_{t=1}^T \mathbb{P}\left[i(t) = i, n_i(t) > N_i, u_i(t) \ge \mu_i + \Delta_i/2, \mathcal{E}_i(t)\right].$$

Picking $N_i = \frac{1280}{\Delta_i^2} \ln\left(\frac{640K}{\Delta_*^2}\right)$ as well as $C_1 = 10$ and $C_2 = 12$, each of these sums can be bounded by application of Lemmas C.2, C.3 and C.4, respectively. We conclude that

$$\sum_{t=1}^T \mathbb{P}\left[i(t) = i\right] \le \frac{1280}{\Delta_i^2} \ln\left(\frac{640K}{\Delta_*^2}\right) + \frac{85}{\Delta_i^2} + \frac{54336}{K} + 16.$$

Plugging this result in the definition of satisficing regret finishes the proof:

$$\mathcal{R}_T^S = \sum_{i:\widetilde{\Delta}_i > 0} \widetilde{\Delta}_i \left(\frac{1280}{\Delta_i^2} \ln\left(\frac{640K}{\Delta_*^2}\right) + \frac{85}{\Delta_i^2} + \frac{54336}{K} + 16\right)$$

$$= \sum_{i:\widetilde{\Delta}_i > 0} \left(1294 \frac{\widetilde{\Delta}_i}{\Delta_i^2} \ln\left(\frac{640K}{\Delta_*^2}\right) + \frac{54336\widetilde{\Delta}_i}{K} + 16\widetilde{\Delta}_i\right),$$

where the last step notices that $85/\ln\left(\frac{640K}{\Delta_*^2}\right) \le 14$ so that the first two terms can be summarized. $\square$

## C.1. Proofs of Lemmas C.2 and C.3

In this section, we provide the proofs of the Lemmas C.2 and C.3. The first lemma follows from the property of the number of uncertain rounds stated in Lemma B.2 and bounds the probability that an insufficient arm $i$ is played without contributing to uncertain rounds too often.

*Proof of Lemma C.2.* Let $i$ be an insufficient arm. By definition of $\mathcal{E}_i(t)$ in Equation (18), $\mathcal{E}_i^c(t) = \{n_{0,i}(t) < n_i(t)/2\} =$

$\{n_{0,i}(t) < n_i(t) - n_i(t)/2\}$, and we have

$$
\begin{aligned}
\sum_{t=1}^{T} \mathbb{P}\left[n_i(t) > N_i, i(t) = i, \ \mathcal{E}_i^c(t)\right] &= \sum_{t=1}^{T} \mathbb{P}\left[n_i(t) > N_i, i(t) = i, n_{0,i}(t) < n_i(t)/2\right] \\
&\leq \sum_{t=1}^{T} \mathbb{P}\left[n_i(t) > N_i, i(t) = i, n_i(t) - n_{0,i}(t) > n_i(t)/2\right] \\
&\leq \sum_{t=1}^{T} \mathbb{P}\left[n_i(t) > N_i, i(t) = i, n_i(T+1) - n_{0,i}(T+1) > n_i(t)/2\right] \qquad (20) \\
&\leq \sum_{n_i=N_i+1}^{T} \mathbb{P}\left[n_i(T+1) - n_{0,i}(T+1) > n_i/2\right] \\
&\leq \sum_{n_i=N_i+1}^{T} \frac{2}{(n_i/2)^{C_2/2-1}} \\
&\leq 16,
\end{aligned}
$$

where Equation (20) notices that $n_i(t) - n_{0,i}(t)$ is an increasing function of $t$, so it is upper bounded by $n_i(T+1) - n_{0,i}(T+1)$. Then, as $n_i(t)$ only increases when $i(t) = i$, we reindex the sum on the number of times arm $i$ is played rather than on $t$. Finally, we apply Lemma B.2 with $n = n_i/2$. For any $C_2 \geq 6$, we have $\sum_{n_i=N_i+1}^{T} \frac{2}{(n_i/2)^{C_2/2-1}} \leq \sum_{n_i=N_i+1}^{T} \frac{2}{(n_i/2)^2} = 8\sum_{n_i=1}^{T} \frac{1}{n_i^2} \leq 16$.

Similarly, when $T \leq \frac{C_2}{4\Delta_*^2}$, we have for any arm $i$ that

$$
\begin{aligned}
\sum_{t=1}^{T} \mathbb{P}\left[n_i(t) > N_i, i(t) = i, \ \mathcal{E}_i^c(t)\right] &\leq \sum_{n_i=N_i+1}^{T} \mathbb{P}\left[n_i(T+1) - n_{0,i}(T+1) > n_i/2\right] \\
&\leq \sum_{n_i=N_i+1}^{T} \frac{8}{(n_i/2)^{C_2/2-1}} \\
&\leq 64,
\end{aligned}
$$

using the same approach and relying on Lemma B.3 with $n = n_i/2$. $\qquad \square$

*Proof of Lemma C.3.* By definition of $\mathcal{E}_i(t)$, we have for any insufficient arm $i$ that

$$
\begin{aligned}
&\sum_{t=1}^{T} \mathbb{P}\left[i(t) = i, n_i(t) > N_i, u_{i^*}(t) \leq \mu_{i^*} - \Delta_i/2, \ \mathcal{E}_i(t)\right] \\
&= \sum_{t=1}^{T} \mathbb{P}\left[i(t) = i, n_i(t) > N_i, \hat{\mu}_{i^*}(t) + \sqrt{C_1 \frac{\ln n_{i^*}(t) + \ln n_0(t)}{n_{i^*}(t)}} \leq \mu_{i^*} - \Delta_i/2, \ n_{0,i}(t) \geq n_i(t)/2\right] \\
&\leq \sum_{t=1}^{T} \mathbb{P}\left[i(t) = i, n_i(t) > N_i, \hat{\mu}_{i^*}(t) + \sqrt{C_1 \frac{\ln n_{i^*}(t) + \ln n_{0,i}(t)}{n_{i^*}(t)}} \leq \mu_{i^*} - \Delta_i/2, \ n_{0,i}(t) \geq n_i(t)/2\right] \\
&\leq \sum_{t=1}^{T} \mathbb{P}\left[i(t) = i, n_i(t) > N_i, \hat{\mu}_{i^*}(t) + \sqrt{C_1 \frac{\ln n_{i^*}(t) + \ln n_i(t)/2}{n_{i^*}(t)}} \leq \mu_{i^*} - \Delta_i/2\right],
\end{aligned}
$$

where the first step replaces $u_{i^*}(t)$ and $\mathcal{E}_i(t)$ by their definitions. The second step lower bounds $n_0(t) \geq n_{0,i}(t)$, and the

third step uses $n_{0,i}(t) \geq n_i(t)/2$. Further simplifications give

$$
\sum_{t=1}^{T} \mathbb{P}\left[ i(t) = i, n_i(t) > N_i, \hat{\mu}_{i^*}(t) + \sqrt{C_1 \frac{\ln n_{i^*}(t) + \ln n_i(t)/2}{n_{i^*}(t)}} \leq \mu_{i^*} - \Delta_i/2 \right]
$$

$$
\leq \sum_{t=1}^{T} \sum_{n_{i^*}=1}^{T} \mathbb{P}\left[ i(t) = i, n_i(t) > N_i, \hat{\mu}_{i^*}(t) = \hat{\mu}_{i^*,n_i^*}, \hat{\mu}_{i^*,n_i^*} + \sqrt{C_1 \frac{\ln n_{i^*} + \ln n_i(t)/2}{n_{i^*}}} \leq \mu_{i^*} - \Delta_i/2 \right]
$$

$$
\leq \sum_{n_i=N_i+1}^{T} \sum_{n_{i^*}=1}^{T} \mathbb{P}\left[ \hat{\mu}_{i^*,n_i^*} + \sqrt{C_1 \frac{\ln n_{i^*} + \ln n_i/2}{n_{i^*}}} \leq \mu_{i^*} - \Delta_i/2 \right]
$$

$$
\leq \sum_{n_i=N_i+1}^{T} \sum_{n_{i^*}=1}^{T} \mathbb{P}\left[ \hat{\mu}_{i^*,n_i^*} \leq \mu_{i^*} - \Delta_i/2 - \sqrt{C_1 \frac{\ln n_{i^*} + \ln n_i/2}{n_{i^*}}} \right]
$$

$$
\leq \sum_{n_i=N_i+1}^{T} \sum_{n_{i^*}=1}^{T} e^{-\frac{n_{i^*}}{2}\left( \Delta_i/2 + \sqrt{C_1 \frac{\ln n_{i^*} + \ln n_i/2}{n_{i^*}}} \right)^2},
$$

where we first take a union bound over all possible values for $n_{i^*}(t)$. In the next step, as the only dependency on $t$ is in $n_i(t)$ and $i(t) = i$, we reindex the sum directly on $n_i$ instead of $t$. Now we can apply our sub-Gaussian assumption as defined in Definition 1, and we obtain

$$
\sum_{n_i=N_i+1}^{T} \sum_{n_{i^*}=1}^{T} e^{-\frac{n_{i^*}}{2}\left( \Delta_i/2 + \sqrt{C_1 \frac{\ln n_{i^*} + \ln n_i/2}{n_{i^*}}} \right)^2} \leq \sum_{n_i=N_i+1}^{T} \sum_{n_{i^*}=1}^{T} e^{-\frac{n_{i^*}}{8}\Delta_i^2 - \frac{C_1}{2}\ln n_{i^*} - \frac{C_1}{2}\ln \frac{n_i}{2}}
$$

$$
\leq \sum_{n_{i^*}=1}^{T} e^{-\frac{n_{i^*}}{8}\Delta_i^2} \sum_{n_i=N_i+1}^{T} e^{-\frac{C_1}{2}\ln \frac{n_i}{2}}
$$

$$
\leq \frac{21}{\Delta_i^2},
$$

where in the first step, we simplify using the fact that $\exp(-x)$ is a decreasing function of $x$ and that $(a+b)^2 \geq a^2 + b^2 \geq a^2$ for $a, b \geq 0$. In the second line, we decompose the result into a product of two convergent sums and discard $e^{-\frac{C_1}{2}\ln n_{i^*}} \leq 1$, as $n_{i^*} \geq 1$ so that $\frac{C_1}{2}\ln n_{i^*} \geq 0$. In that second line, the first sum is bounded by $\frac{8}{\Delta_i^2}$ and the second can be loosely bounded as $\sum_{n_i=N_i+1}^{T} e^{-\frac{C_1}{2}\ln \frac{n_i}{2}} = \sum_{n_i=N_i+1}^{T} \left(\frac{2}{n_i}\right)^{C_1/2} \leq \sum_{n_i=2}^{T} \left(\frac{2}{n_i}\right)^2 \leq 2.6$, provided that $C_1 \geq 4$. (Note that we are summing from $n_i = 2$, so that $0 < \frac{2}{n_i} \leq 1$ for each $n_i$ and thus we take an upper bound by lower bounding the exponent.)

We conclude that for any $N_i \geq 1$ we have

$$
\sum_{t=1}^{T} \mathbb{P}\left[ i(t) = i, n_i(t) > N_i, u_{i^*}(t) \leq \mu_{i^*} - \Delta_i/2, \mathcal{E}_i(t) \right] \leq \frac{21}{\Delta_i^2}.
$$

$\square$

### C.2. Proof of Lemma C.4

The main challenge in the analysis of Equation (19) comes with bounding

$$
\sum_{t=1}^{T} \mathbb{P}\left[ i(t) = i, n_i(t) > N_i, u_i(t) \geq \mu_i + \Delta_i/2, \mathcal{E}_i(t) \right],
$$

as it requires further decomposition. We introduce a new event around which we will decompose the sum:

$$
M(t, n) := \left\{ \max_{j \in [K]} n_{0,j}(t) \leq n \right\} \tag{21}
$$

Then, the last sum in Equation (19) can be decomposed into

$$\sum_{t=1}^{T} \mathbb{P}\left[i(t) = i, n_i(t) > N_i, u_i(t) \geq \mu_i + \Delta_i/2, \ \mathcal{E}_i(t)\right]$$

$$\leq \sum_{t=1}^{T} \mathbb{P}\left[i(t) = i, n_i(t) > N_i, u_i(t) \geq \mu_i + \Delta_i/2, \ \mathcal{E}_i(t), M(t, n)\right] \tag{22}$$

$$+ \sum_{t=1}^{T} \mathbb{P}\left[i(t) = i, n_i(t) > N_i, u_i(t) \geq \mu_i + \Delta_i/2, \ \mathcal{E}_i(t), M^c(t, n)\right]. \tag{23}$$

By definition of $M(t, n)$, we have $M^c(t, n) = \left\{\max_{j \in [K]} n_{0,j}(t) > n\right\}$, and we need $n$ to represent some measure of how challenging the problem instance is. To do so, we decompose our set of arms into two categories.

**Definition C.5.** For any $K$-armed satisficing bandit problem, we can partition the arms into two sets:

$$\mathcal{V} := \{i \in [K] : \Delta_i \leq \Delta_*/2\}$$
$$\mathcal{V}^c := \{i \in [K] : \Delta_i > \Delta_*/2\}$$

The set $\mathcal{V}$ denotes the "very good" arms, which are the arms that are not only sufficient, but that have an expected reward closer to $\mu_*$ than to $S$. For these arms, we want to show that $n_{0,i}(s)$ is small even though they might be played often (meaning that $n_i(s)$ could be large), because they don't remain uncertain for many rounds. For the other arms in the set $\mathcal{V}^c$ we show that they are not played very often so $n_{0,i}(s) \leq n_i(s)$ does not get too large.

We set $N_{0\max} = \frac{64}{\Delta_*^2}$ so that $N_{0\max} \geq \max\left\{\max_{j \in \mathcal{V}} \frac{16}{\Delta_j^2}, \max_{j \in \mathcal{V}^c} \frac{64}{\Delta_j^2}\right\}$. We want $n$ to be proportional to $N_{0\max}$ in Equations (22) and (23). Specifically, we set $n = L N_{0\max}$ for a certain constant $L \geq 1$, and can decompose Equation (23) by considering how far $\max_j n_{0,j}(t)$ is above $N_{0\max}$: For any $t$ with $\max_j n_{0,j}(t) > L N_{0\max}$, there exists $\ell \geq L$ such that $(\ell + 1) N_{0\max} \geq \max_j n_{0,j}(t) > \ell N_{0\max}$. Then, we can decompose Equation (23) by partitioning and summing over $\ell$:

$$\sum_{t=1}^{T} \mathbb{P}\left[i(t) = i, n_i(t) > N_i, u_i(t) \geq \mu_i + \Delta_i/2, \ \mathcal{E}_i(t), M^c(t, L N_{0\max})\right]$$

$$\leq \sum_{t=1}^{T} \sum_{\ell \geq L} \mathbb{P}\left[i(t) = i, n_i(t) > N_i, u_i(t) \geq \mu_i + \Delta_i/2, \ \mathcal{E}_i(t), M^c(t, \ell N_{0\max}), M(t, (\ell + 1) N_{0\max})\right]$$

$$\leq \sum_{t=1}^{T} \sum_{\ell \geq L : n_{0,i}(t) \geq (\ell+1) N_{0\max}} \mathbb{P}\Big[i(t) = i, n_i(t) > N_i, u_i(t) \geq \mu_i + \Delta_i/2, \ \mathcal{E}_i(t),$$

$$M^c(t, \ell N_{0\max}), M(t, (\ell + 1) N_{0\max})\Big] \tag{24}$$

$$+ \sum_{t=1}^{T} \sum_{\ell \geq L : n_{0,i}(t) < (\ell+1) N_{0\max}} \mathbb{P}\Big[i(t)i, n_i(t) > N_i, u_i(t) \geq \mu_i + \Delta_i/2, \ \mathcal{E}_i(t),$$

$$M^c(t, \ell N_{0\max}), M(t, (\ell + 1) N_{0\max})\Big]. \tag{25}$$

Equations (22), (24), and (25) can be bounded according to the three following lemmas.

**Lemma C.6.** *Consider any sub-Gaussian $K$-armed bandit problem. Using* uncertain-UCB *with $C_1 \geq 4$ and $C_2 \geq 6$, for any insufficient arm $i$, any $L$ and any $N_{0max} \geq 1$, if $N_i \geq \frac{32 C_1}{\Delta_i^2} \max\left\{\ln\left(K L N_{0max}\right), 2\ln\left(\frac{32 C_1}{\Delta_i^2}\right)\right\}$, then*

$$\sum_{t=1}^{T} \mathbb{P}\left[i(t) = i, n_i(t) > N_i, u_i(t) \geq \mu_i + \Delta_i/2, \ \mathcal{E}_i(t), M(t, \ell N_{0max})\right] \leq \frac{32}{\Delta_i^2}.$$

**Lemma C.7.** *Consider any sub-Gaussian $K$-armed bandit problem. For any insufficient arm $i$, using* uncertain-UCB *with $C_1 \geq 4$ and $C_2 \geq 6$, if $N_i \geq \frac{128C_1}{\Delta_i^2} \ln\left(\frac{64C_1 K}{\Delta_i^2}\right)$ then it holds for any $L$ and any $N_{0max} \geq 1$ that*

$$\sum_{t=1}^{T} \sum_{\ell \geq L : n_{0,i}(t) \geq (\ell+1) N_{0max}} \mathbb{P}\big[i(t) = i, n_i(t) > N_i, u_i(t) \geq \mu_i + \Delta_i/2,$$

$$\mathcal{E}_i(t), M^c(t, \ell N_{0max}), M(t, (\ell+1) N_{0max})\big] \leq \frac{32}{\Delta_i^2}.$$

**Lemma C.8.** *Consider any sub-Gaussian $K$-armed bandit problem. For any insufficient arm $i$, using* uncertain-UCB *with $C_1 = 10$ and $C_2 = 12$, if $N_i \geq 1$ then for any $N_{0max} \geq \frac{64}{\Delta_*^2}$ and $L \geq \max\left\{K/\Delta_*^2, 3\right\}$ we have*

$$\sum_{t=1}^{T} \sum_{\ell \geq L : n_{0,i}(t) < (\ell+1) N_{0max}} \mathbb{P}\big[i(t)i, n_i(t) > N_i, u_i(t) \geq \mu_i + \Delta_i/2,$$

$$\mathcal{E}_i(t), M^c(t, \ell N_{0max}), M(t, (\ell+1) N_{0max})\big]$$

$$\leq \left(816 + 64L^2 \frac{3(L-1)+1}{6(L-2)^2}\right) \frac{N_{0max} \Delta_*^2}{K}.$$

With all these results established, we can move on to the proof of Lemma C.4.

*Proof of Lemma C.4.* We use Lemmas C.6, C.7 and C.8 to bound Equations (22), (24), and (25), respectively. To apply these results simultaneously, we need to ensure that $N_{0max}$, $N_i$ and $L$ are large enough. Setting $N_{0max} = \frac{64}{\Delta_*^2}$ and recalling that $C_1 = 10$ and $C_2 = 12$, we note that the constraint on $N_i$ required for Lemma C.6 to hold is more limiting than the constraint required for Lemma C.7. This means that we can pick a large value for $L$ such as $L = \frac{6400K}{\Delta_*^2}$, and then set $N_i = \frac{1280}{\Delta_i^2} \ln\left(\frac{640K}{\Delta_*^2}\right)$ to satisfy all the conditions needed to apply Lemmas C.6, C.7 and C.8. Moreover, as the bound coming from Lemma C.8 depends on $L$, we note that $L = \frac{6400K}{\Delta_*^2} \geq 6400$, so that $L^2 \frac{3(L-1)+1}{6(L-2)^2} \leq 0.51$.

Then summing the bounds of Lemma C.6, Lemma C.7 and Lemma C.8 we obtain that

$$\sum_{t=1}^{T} \mathbb{P}\left[n_i(t) > N_i, i(t) = i, u_i(t) \geq \mu_i + \Delta_i/2, \mathcal{E}_i(t)\right] \leq \frac{64}{\Delta_i^2} + 849 \frac{N_{0max} \Delta_*^2}{K}$$

$$\leq \frac{64}{\Delta_i^2} + \frac{54336}{K}.$$

$\square$

Let us now move on to the proofs of Lemmas C.6, C.7, and C.8.

*Proof of Lemma C.6.* Using the definition of $M(t, L N_{0max})$, we note that we are in the case where $\max_{j \in [K]} n_{0,j}(t) \leq L N_{0max}$, which in particular implies that

$$n_0(t) \leq K \max_{j \in [K]} n_{0,j}(t) \leq K L N_{0max}.$$

Then we can bound our expression as

$$\sum_{t=1}^{T} \mathbb{P}\left[i(t) = i, n_i(t) > N_i, u_i(t) \geq \mu_i + \Delta_i/2, \ \mathcal{E}_i(t), M(t, LN_{0\max})\right]$$

$$\leq \sum_{t=1}^{T} \mathbb{P}\left[i(t) = i, n_i(t) > N_i, \hat{\mu}_i(t) + \sqrt{C_1 \frac{\ln n_i(t) + \ln n_0(t)}{n_i(t)}} \geq \mu_i + \Delta_i/2, M(t, LN_{0\max})\right]$$

$$\leq \sum_{t=1}^{T} \mathbb{P}\left[i(t) = i, n_i(t) > N_i, \hat{\mu}_i(t) + \sqrt{C_1 \frac{\ln n_i(t) + \ln KLN_{0\max}}{n_i(t)}} \geq \mu_i + \Delta_i/2\right]$$

$$\leq \sum_{n_i \geq N_i} \mathbb{P}\left[\hat{\mu}_{i,n_i} + \sqrt{C_1 \frac{\ln n_i + \ln KLN_{0\max}}{n_i}} \geq \mu_i + \Delta_i/2\right], \tag{26}$$

where the first step replaces $u_i(t)$ by its definition and drops the condition on $\mathcal{E}_i(t)$. The second step replaces $n_0(t)$ by its bound using the condition $M(t, LN_{0\max})$. We then reuse the approach of reindexing the sum on $t$ by a sum on $n_i$ as we only sum on rounds where arm $i$ is played, when $n_i$ increases.

To use the sub-Gaussianity assumption, we have to guarantee that $\Delta_i/2 - \sqrt{C_1 \frac{\ln n_i + \ln KLN_{0\max}}{n_i}} > 0$ for all $n_i \geq N_i$. This is guaranteed by Lemma C.11 from Appendix C.3 below, which implies that when choosing $N_i \geq \frac{32C_1}{\Delta_i^2} \max\left\{\ln(KLN_{0\max}), 2\ln\left(\frac{32C_1}{\Delta_i^2}\right)\right\}$ then for all $n_i \geq N_i$ it holds that $\sqrt{C_1 \frac{\ln n_i + \ln KLN_{0\max}}{n_i}} \leq \Delta_i/4$, which in turn implies

$$\Delta_i/2 - \sqrt{C_1 \frac{\ln n_i + \ln KLN_{0\max}}{n_i}} \geq \Delta_i/4 > 0.$$

It follows that

$$\sum_{t=1}^{T} \mathbb{P}\left[i(t) = i, n_i(t) > N_i, u_i(t) \geq \mu_i + \Delta_i/2, \ \mathcal{E}_i(t), M(t, LN_{0\max})\right]$$

$$\leq \sum_{n_i \geq N_i} \mathbb{P}\left[\hat{\mu}_{i,n_i} - \mu_i \geq \Delta_i/4\right]$$

$$\leq \sum_{n_i \geq N_i} e^{-\frac{n_i \Delta_i^2}{32}}$$

$$\leq \frac{32}{\Delta_i^2},$$

where we rely on the sub-Gaussianity to ensure that $\mathbb{P}\left[\hat{\mu}_{i,n_i} - \mu_i \geq \varepsilon\right] \leq e^{-n_i \varepsilon^2/2}$ when $\varepsilon > 0$ and then use $\sum_{n_i \geq 1} \exp(-n_i c) \leq 1/c$ for $c > 0$. $\qquad\square$

*Proof of Lemma C.7.* We aim to bound

$$\sum_{t=1}^{T} \sum_{\ell \geq L : n_{0,i}(t) \geq (\ell+1)N_{0\max}} \mathbb{P}\Big[i(t) = i, n_i(t) > N_i, u_i(t) \geq \mu_i + \Delta_i/2, \ \mathcal{E}_i(t),$$

$$M^c(t, \ell N_{0\max}), M(t, (\ell+1)N_{0\max})\Big]. \tag{27}$$

We note that by definition of $n_0(t)$ and $M(t, (\ell+1)N_{0\max})$, we have

$$n_0(t) = \sum_{j=1}^{K} n_{0,j}(t) \leq K \max_{j \in [K]} n_{0,j}(t) \leq K(\ell+1)N_{0\max}.$$

We deduce that $n_0(t) \leq K(\ell+1)N_{0\max} \leq Kn_{0,i}(t) \leq Kn_i(t)$, as we are summing on values of $\ell$ such that $n_{0,i}(t) \geq (\ell+1)N_{0\max}$. Furthermore, we are only summing on terms with $n_{0,i}(t) \geq (\ell+1)N_{0\max}$, but under $M(t, (\ell+1)N_{0\max})$ we

also have that $n_{0,i}(t) \leq \max_j n_{0,j}(t) \leq (\ell+1)N_{0\max}$, so the inner sum in Equation (27) has at most one term, corresponding to $n_{0,i}(t) = (\ell+1)N_{0\max}$. Removing the unnecessary conditions, we are left to bound

$$\sum_{t=1}^{T} \sum_{\ell \geq L:n_{0,i}(t) \geq (\ell+1)N_{0\max}} \mathbb{P}\big[i(t) = i, n_i(t) > N_i, u_i(t) \geq \mu_i + \Delta_i/2, \, \mathcal{E}_i(t),$$

$$M^c(t, \ell N_{0\max}), M(t, (\ell+1)N_{0\max})\big]$$

$$\leq \sum_{t=1}^{T} \mathbb{P}\left[i(t) = i, n_i(t) > N_i, u_i(t) \geq \mu_i + \Delta_i/2\right]$$

$$\leq \sum_{t=1}^{T} \mathbb{P}\left[i(t) = i, n_i(t) > N_i, \hat{\mu}_i(t) + \sqrt{C_1 \frac{\ln n_i(t) + \ln K n_i(t)}{n_i(t)}} \geq \mu_i + \Delta_i/2\right]$$

$$\leq \sum_{n_i \geq N_i} \mathbb{P}\left[\hat{\mu}_{i,n_i} + \sqrt{C_1 \frac{\ln n_i + \ln K n_i}{n_i}} \geq \mu_i + \Delta_i/2\right]$$

$$\leq \sum_{n_i \geq N_i} \mathbb{P}\left[\hat{\mu}_{i,n_i} + \sqrt{2C_1 \frac{\ln K n_i}{n_i}} \geq \mu_i + \Delta_i/2\right], \tag{28}$$

where we used the reindexing trick on the third inequality, as the only dependence on $t$ is in $i(t)$ and $n_i(t)$, and $n_i(t)$ only increases when arm $i$ is played. Then, we can use that $\hat{\mu}_{i,n_i}$ is sub-Gaussian and using Lemma C.11, we conclude that when $N_i \geq \frac{128 C_1}{\Delta_i^2} \ln\left(\frac{64 C_1 K}{\Delta_i^2}\right)$, we have that for all $n_i \geq N_i$,

$$\sqrt{2C_1 \frac{\ln K n_i}{n_i}} = \sqrt{2C_1 \frac{\ln n_i + \ln K}{n_i}} \leq \Delta_i/4, \text{ and so } \frac{\Delta_i}{2} - \sqrt{2C_1 \frac{\ln K n_i}{n_i}} \geq \Delta_i/4 > 0,$$

where the last step follows from $i \in \mathcal{V}^c$, so $\Delta_i > 0$. Then Equation (28) can be bounded as

$$\sum_{n_i \geq N_i} \mathbb{P}\left[\hat{\mu}_{i,n_i} + \sqrt{2C_1 \frac{\ln K n_i}{n_i}} \geq \mu_i + \Delta_i/2\right] \leq \sum_{n_i \geq N_i} \mathbb{P}\left[\hat{\mu}_{i,n_i} \geq \mu_i - \Delta_i/2 + \sqrt{2C_1 \frac{\ln K n_i}{n_i}}\right]$$

$$\leq \sum_{n_i \geq N_i} \mathbb{P}\left[\hat{\mu}_{i,n_i} - \mu_i \geq \Delta_i/4\right]$$

$$\leq \sum_{n_i \geq N_i} e^{-\frac{n_i \Delta_i^2}{32}}$$

$$\leq \frac{32}{\Delta_i^2},$$

where we use as at the end of the proof of Lemma C.6 that $\mathbb{P}\left[\hat{\mu}_{i,n_i} - \mu_i \geq \varepsilon\right] \leq e^{-n_i \varepsilon^2/2}$ for $\varepsilon > 0$ and $\sum_{n_i \geq 1} \exp(-n_i c) \leq 1/c$ for $c > 0$. $\qquad \square$

*Proof of Lemma C.8.* We want to bound

$$\sum_{t=1}^{T} \sum_{\ell \geq L:n_{0,i}(t) < (\ell+1)N_{0\max}} \mathbb{P}\big[i(t) = i, n_i(t) > N_i, u_i(t) \geq \mu_i + \Delta_i/2, \, \mathcal{E}_i(t),$$

$$M^c(t, \ell N_{0\max}), M(t, (\ell+1)N_{0\max})\big]. \tag{29}$$

To do so, we first recall the definition of $\mathcal{E}_i(t) = \{n_{0,i}(t) \geq n_i(t)/2\}$. This means that in the indexing of the sum, when we verify that $n_{0,i}(t) < (\ell+1)N_{0\max}$, we can lower bound this quantity using $n_i(t)/2 \leq n_{0,i}(t) \leq (\ell+1)N_{0\max}$, so that we

can bound Equation (29) by

$$
\sum_{t=1}^{T} \sum_{\ell \geq L: n_i(t) < 2(\ell+1)N_{0\text{max}}} \mathbb{P}\big[i(t) = i, n_i(t) > N_i, u_i(t) \geq \mu_i + \Delta_i/2,\ \mathcal{E}_i(t),
$$
$$
M^c(t, \ell N_{0\text{max}}), M(t, (\ell+1)N_{0\text{max}})\big]
$$
$$
\leq \sum_{t=1}^{T} \sum_{\ell \geq L: n_i(t) < 2(\ell+1)N_{0\text{max}}} \mathbb{P}\left[i(t) = i, n_i(t) > N_i, M^c(t, \ell N_{0\text{max}}), M(t, (\ell+1)N_{0\text{max}})\right]
$$
$$
\leq \sum_{n_i \geq N_i} \sum_{\ell \geq L: n_i < 2(\ell+1)N_{0\text{max}}} \mathbb{P}\left[\max_{j \in [K]} n_{0,j}(t) > \ell N_{0\text{max}}\right]
$$
$$
\leq \sum_{\ell \geq L} \sum_{n_i = N_i}^{2(\ell+1)N_{0\text{max}}} \mathbb{P}\left[\max_{j \in [K]} n_{0,j}(t) > \ell N_{0\text{max}}\right],
$$

where in the second step we again reindex the sum on $t$ to a sum on $n_i$. We then swap the sums so that both constraints on $n_i$ are highlighted in the second sum. Using Lemma C.10 below with $n = \ell N_{0\text{max}}$, and picking $C_1 = 10$ and $C_2 = 12$, we have

$$
\sum_{\ell \geq L} \sum_{n_i = N_i}^{2(\ell+1)N_{0\text{max}}} \mathbb{P}\left[\max_{j \in [K]} n_{0,j} > \ell N_{0\text{max}}\right]
$$
$$
\leq \sum_{\ell \geq L} \sum_{n_i = N_i}^{2(\ell+1)N_{0\text{max}}} \left(\sum_{j \in \mathcal{V}} \frac{8}{\hat{\Delta}_j^2} e^{-\frac{\ell N_{0\text{max}} \hat{\Delta}_j^2}{16}} + \sum_{j \in \mathcal{V}^c} \left(\frac{32}{\Delta_j^2} e^{-\frac{\ell N_{0\text{max}} \Delta_j^2}{64}} + \frac{32}{(\ell N_{0\text{max}} - 1)^4}\right)\right)
$$
$$
\leq \sum_{j \in \mathcal{V}} \frac{16 N_{0\text{max}}}{\hat{\Delta}_j^2} \sum_{\ell \geq L} (\ell+1) e^{-\frac{\ell N_{0\text{max}} \hat{\Delta}_j^2}{16}} + \sum_{j \in \mathcal{V}^c} \frac{64 N_{0\text{max}}}{\Delta_j^2} \sum_{\ell \geq L} (\ell+1) e^{-\frac{\ell N_{0\text{max}} \Delta_j^2}{64}}
$$
$$
+ 64 N_{0\text{max}} \sum_{j \in \mathcal{V}^c} \sum_{\ell \geq L} \frac{\ell+1}{(\ell N_{0\text{max}} - 1)^4}, \tag{30}
$$

where we note that the terms in the sum $\sum_{n_i = N_i}^{2(\ell+1)N_{0\text{max}}}$ are independent of $n_i$, so we can just bound them by the number of terms.

We now bound the first two sums in Equation (30). By definition, for any $j \in \mathcal{V}$ we have $N_{0\text{max}} \geq 16/\hat{\Delta}_j^2$ and $\hat{\Delta}_j \geq \Delta_*/2$. Similarly, for any $j \in \mathcal{V}^c$ it holds that $N_{0\text{max}} \geq 64/\Delta_j^2$ and $\Delta_j \geq \Delta_*/2$. Then, for any $L \geq \frac{K}{\Delta_*^2} \geq 1$, we recall that $\exp(-x)$ is a decreasing function of $x$ and we obtain

$$
\sum_{j \in \mathcal{V}} \frac{16 N_{0\text{max}}}{\hat{\Delta}_j^2} \sum_{\ell \geq L} (\ell+1) e^{-\frac{\ell N_{0\text{max}} \hat{\Delta}_j^2}{16}} + \sum_{j \in \mathcal{V}^c} \frac{64 N_{0\text{max}}}{\Delta_j^2} \sum_{\ell \geq L} (\ell+1) e^{-\frac{\ell N_{0\text{max}} \Delta_j^2}{64}}
$$
$$
\leq \sum_{j \in [K]} \frac{272 N_{0\text{max}}}{\Delta_*^2} \sum_{\ell \geq L} (\ell+1) e^{-\frac{\ell N_{0\text{max}} \Delta_*^2}{64}}
$$
$$
\leq K \frac{272 N_{0\text{max}}}{\Delta_*^2} \sum_{\ell \geq L} (\ell+1) e^{-\frac{\ell N_{0\text{max}} \Delta_*^2}{64}}
$$
$$
\leq \frac{272 N_{0\text{max}} \Delta_*^2}{K} L^2 \sum_{\ell \geq L} (\ell+1) e^{-\frac{\ell N_{0\text{max}} \Delta_*^2}{64}}
$$
$$
\leq \frac{816 N_{0\text{max}} \Delta_*^2}{K},
$$

where we first upper bound the sum on $j$ by a factor $K$. At the third step, we use that $\frac{K}{\Delta_*^2} \leq L$ and hence $\frac{K^2}{\Delta_*^4} \leq L^2$. In the last step, we apply Lemma C.9 below, noting that $N_{0\text{max}} \Delta_*^2/64 \geq 1$.

For the third sum in Equation (30), we first note that $\frac{\ell+1}{(\ell N_{0\max}-1)^4}$ is decreasing in $N_{0\max}$, so it holds that $\frac{\ell+1}{(\ell N_{0\max}-1)^4} \leq \frac{\ell+1}{(\ell-1)^4}$ since $N_{0\max} \geq 1$. Using that $L \geq K/\Delta_*^2$ and $|\mathcal{V}^c| \leq K$, we obtain

$$64N_{0\max} \sum_{j\in\mathcal{V}^c}\sum_{\ell\geq L} \frac{\ell+1}{(\ell N_{0\max}-1)^4} \leq 64N_{0\max}K\sum_{\ell\geq L}\frac{\ell+1}{(\ell-1)^4} \leq 64N_{0\max}\frac{\Delta_*^2}{K}L^2\sum_{\ell\geq L}\frac{\ell+1}{(\ell-1)^4}.$$

Then, using that $\frac{\ell+1}{(\ell-1)^4}$ is a decreasing function of $\ell$, we get an upper bound of

$$L^2\sum_{\ell\geq L}\frac{\ell+1}{(\ell-1)^4} \leq L^2\int_{L-1}^{\infty}\frac{\ell+1}{(\ell-1)^4}d\ell \leq L^2\frac{3(L-1)+1}{6(L-2)^3}.$$

Bringing these results together, we deduce that

$$\sum_{\ell\geq L}\sum_{n_i=N_i}^{2(\ell+1)N_{0\max}} \mathbb{P}\left[\max_{j\in[K]}n_{0,j}>\ell N_{0\max}\right] \leq \left(816+64L^2\frac{3(L-1)+1}{6(L-2)^2}\right)\frac{N_{0\max}\Delta_*^2}{K}.$$

$\square$

**Lemma C.9.** *If $L \geq 1$ and $c \geq 1$ then*

$$L^2\sum_{\ell\geq L}(\ell+1)e^{-c\ell} \leq 3.$$

*Proof of Lemma C.9.* As $\exp(-cx)$ is decreasing in $c$, we have $\sum_{\ell\geq L}(\ell+1)e^{-c\ell} \leq \sum_{\ell\geq L}(\ell+1)e^{-\ell}$ for $c \geq 1$. It follows that

$$\sum_{\ell\geq L}(\ell+1)e^{-\ell} \leq \int_{L-1}^{\infty}(\ell+1)e^{-\ell}d\ell \leq (L+1)e^{-(L-1)}.$$

and hence for any $L \geq 1$,

$$L^2\sum_{\ell\geq L}(\ell+1)e^{-c\ell} \leq L^2(L+1)e^{-(L-1)} \leq 3.$$

$\square$

**Lemma C.10.** *Under the conditions of Lemma C.8, we have for any $n \geq LN_{0max}$,*

$$\mathbb{P}\left[\max_{j\in[K]}n_{0,j}(t)>n\right] \leq \sum_{i\in\mathcal{V}}\frac{8}{\hat{\Delta}_i^2}e^{-\frac{n\hat{\Delta}_i^2}{16}} + \sum_{i\in\mathcal{V}^c}\frac{32}{\Delta_i^2}e^{-\frac{n\Delta_i^2}{64}}$$

$$+ \sum_{i\in\mathcal{V}^c}\frac{2^{C_1/2+1}}{C_1/2-1}\cdot\frac{1}{(n-1)^{C_1/2-1}} + \sum_{i\in\mathcal{V}^c}\frac{2^{C_2/2}}{(C_2/2-2)}\cdot\frac{1}{(n-1)^{C_2/2-2}}.$$

*In particular, for $C_1 = 10$ and $C_2 = 12$ it holds that*

$$\mathbb{P}\left[\max_{j\in[K]}n_{0,j}(t)>n\right] \leq \sum_{i\in\mathcal{V}}\frac{8}{\hat{\Delta}_i^2}e^{-\frac{n\hat{\Delta}_i^2}{16}} + \sum_{i\in\mathcal{V}^c}\left(\frac{32}{\Delta_i^2}e^{-\frac{n\Delta_i^2}{64}}+\frac{32}{(n-1)^4}\right).$$

*Proof of Lemma C.10.* We want to bound the term $\mathbb{P}\left[\max_{j\in[K]}n_{0,j}(t)>n\right]$ for some fixed $n$. We can decompose this expression as follows. Let $\mathcal{U}(s) = \text{argmax}_j n_{0,j}(s)$ denote the set of arms that have contributed most to $n_0(s)$ at round $s$. We note that if $\max_j n_{0,j}(t)>n$ for some round $t$, then there exists a round $s \leq t$ where that condition first failed, that is, a round $s \leq t$ such that $\max_j n_{0,j}(s-1) = n$ and $\max_j n_{0,j}(s) = n+1$. Then

$$\mathbb{P}\left[\max_{j\in[K]}n_{0,j}(t)>n\right] \leq \mathbb{P}\left[\exists s < t : i(s) = i, i \in \mathcal{U}(s), \max_{j\in[K]}n_{0,j}(s) = n, n_{0,i}(s+1) = n+1\right]$$

$$\leq \sum_{s=n}^{t}\mathbb{P}\left[i(s) = i, i \in \mathcal{U}(s), \max_{j\in[K]}n_{0,j}(s) = n, n_{0,i}(s+1) = n+1\right]. \qquad (31)$$

Using the decomposition of the arms into the sets $\mathcal{V}$ and $\mathcal{V}^c$ according to Definition C.5, Equation (31) can be bounded as

$$\sum_{s=n}^{t} \mathbb{P}\left[i(s) = i, i \in \mathcal{U}(s), \max_{j \in [K]} n_{0,j}(s) = n, n_{0,i}(s+1) = n+1\right]$$

$$\leq \sum_{s=n}^{t} \sum_{i \in \mathcal{V}} \mathbb{P}\left[i(s) = i, i \in \mathcal{U}(s), \max_{j \in [K]} n_{0,j}(s) = n, n_{0,i}(s+1) = n+1\right]$$

$$+ \sum_{s=n}^{t} \sum_{i \in \mathcal{V}^c} \mathbb{P}\left[i(s) = i, i \in \mathcal{U}(s), \max_{j \in [K]} n_{0,j}(s) = n, n_{0,i}(s+1) = n+1\right]$$

$$\leq \sum_{s=n}^{t} \sum_{i \in \mathcal{V}} \mathbb{P}\left[i(s) = i, i \in \mathcal{U}(s), \max_{j \in [K]} n_{0,j}(s) = n, \hat{\mu}_i(s) < S + \sqrt{C_2 \frac{\ln n_i(s)}{n_i(s)}}\right] \tag{32}$$

$$+ \sum_{s=n}^{t} \sum_{i \in \mathcal{V}^c} \mathbb{P}\left[i(s) = i, i \in \mathcal{U}(s), \max_{j \in [K]} n_{0,j}(s) = n, u_i(s) \geq \mu_i + \Delta_i/2\right] \tag{33}$$

$$+ \sum_{s=n}^{t} \sum_{i \in \mathcal{V}^c} \mathbb{P}\left[i(s) = i, i \in \mathcal{U}(s), \max_{j \in [K]} n_{0,j}(s) = n, u_{i^*}(s) \leq \mu_{i^*} - \Delta_i/2\right], \tag{34}$$

where in the last step we use that for the arms in $\mathcal{V}$, the probability that $n_{0,i}$ increased between step $s$ and $s+1$ can be rewritten according to the definition of $n_0$ in Equation (16). For the other arms, we bound $i(s) = i$ by noticing that when arm $i \in \mathcal{V}^c$ is played, it means that either $u_i(s)$ is overestimated or $u_{i^*}(s)$ is underestimated.

We now are going to handle each of the three terms separately.

**Bounding Equation (32)**  This first term is bounded as

$$\sum_{s=n}^{t} \sum_{i \in \mathcal{V}} \mathbb{P}\left[i(s) = i, i \in \mathcal{U}(s), \max_{j \in [K]} n_{0,j}(s) = n, \hat{\mu}_i(s) < S + \sqrt{C_2 \frac{\ln n_i(s)}{n_i(s)}}\right]$$

$$\leq \sum_{i \in \mathcal{V}} \sum_{n_i \geq n} \mathbb{P}\left[\hat{\mu}_{i,n_i} < S + \sqrt{C_2 \frac{\ln n_i}{n_i}}\right]$$

$$\leq \sum_{i \in \mathcal{V}} \sum_{n_i \geq n} \mathbb{P}\left[\hat{\mu}_{i,n_i} < \mu_i - \hat{\Delta}_i + \sqrt{C_2 \frac{\ln n_i}{n_i}}\right], \tag{35}$$

where the first step replaces dependencies on $t$ by a dependency on $n_i$. It is sufficient to sum on $n_i \geq n$ as for any $s$ in the original sum, $n_{0,i}(s) = n$ and $n_{0,i}(s) \leq n_i(s)$. In the second step, we use that $\mu_i > S$ so $S = \mu_i - \hat{\Delta}_i$. Noting that $\hat{\Delta}_i \geq \Delta_{i^*}/2$ due to $i \in \mathcal{V}$, it holds that $n \geq LN_{0\max} \geq \frac{16 C_2}{\hat{\Delta}_i^2} \ln\left(\frac{16 C_2}{\hat{\Delta}_i^2}\right)$, and we have by Lemma C.11 below that $\sqrt{C_2 \frac{\ln n_i}{n_i}} \leq \hat{\Delta}_i/2$ for all $n_i \geq n$, which also guarantees that $\hat{\Delta}_i - \sqrt{C_2 \frac{\ln n_i}{n_i}} > \hat{\Delta}_i/2 > 0$. Accordingly, we can use that the rewards are sub-Gaussian to get a bound Equation (35):

$$\sum_{i \in \mathcal{V}} \sum_{n_i \geq n} \mathbb{P}\left[\hat{\mu}_{i,n_i} < \mu_i - \hat{\Delta}_i + \sqrt{C_2 \frac{\ln n_i}{n_i}}\right] \leq \sum_{i \in \mathcal{V}} \sum_{n_i \geq n} \mathbb{P}\left[\hat{\mu}_{i,n_i} - \mu_i \geq \hat{\Delta}_i/2\right]$$

$$\leq \sum_{i \in \mathcal{V}} \sum_{n_i \geq n} e^{-\frac{n_i \hat{\Delta}_i^2}{8}}$$

$$\leq \sum_{i \in \mathcal{V}} \frac{8}{\hat{\Delta}_i^2} e^{-\frac{n \hat{\Delta}_i^2}{16}},$$

where once more we have $\mathbb{P}[\hat{\mu}_{i,n_i} - \mu_i \geq \varepsilon] \leq e^{-n_i \varepsilon^2/2}$ for $\varepsilon > 0$ due to sub-Gaussianity and we further use that $\sum_{n_i \geq n} \exp(-n_i c) \leq \frac{e^{-cn/2}}{c}$ for $c > 0$.

**Bounding Equation (33)** This term is bounded using the same approach as in the proof of Lemma C.6. We start with

$$\sum_{s=n}^{t}\sum_{i\in\mathcal{V}^c}\mathbb{P}\left[i(s)=i, i\in\mathcal{U}(s), \max_{j\in[K]}n_{0,j}(s)=n, n_{0,i}(s+1)=n+1, u_i(s)\geq\mu_i+\Delta_i/2\right]$$

$$=\sum_{s=n}^{t}\sum_{i\in\mathcal{V}^c}\mathbb{P}\left[i(s)=i, i\in\mathcal{U}(s), \max_{j\in[K]}n_{0,j}(s)=n, \hat{\mu}_i(s)+\sqrt{C_1\frac{\ln n_i(s)+\ln n_0(s)}{n_i(s)}}\geq\mu_i+\Delta_i/2\right]$$

$$\leq\sum_{s=n}^{t}\sum_{i\in\mathcal{V}^c}\mathbb{P}\left[i(s)=i, i\in\mathcal{U}(s), \max_{j\in[K]}n_{0,j}(s)=n, \hat{\mu}_i(s)+\sqrt{C_1\frac{\ln n_i(s)+\ln Kn}{n_i(s)}}\geq\mu_i+\Delta_i/2\right]$$

$$\leq\sum_{i\in\mathcal{V}^c}\sum_{n_i\geq n}\mathbb{P}\left[\hat{\mu}_{i,n_i}+\sqrt{C_1\frac{\ln n_i+\ln Kn}{n_i}}\geq\mu_i+\Delta_i/2\right], \tag{36}$$

where the first step replaces $u_i(s)$ by its definition, the second step replaces $n_0(s)\leq K\max_j n_0(s)=Kn$, and the third step reindexes the sum on $n_i$ rather than $t$.

Now we use that $i\in\mathcal{V}^c$ so that $\Delta_i\geq\Delta_{i*}/2$, and it follows that $n\geq LN_{0\max}\geq\frac{32C_1}{\Delta_i^2}\max\left\{\ln(Kn), 2\ln\left(\frac{32C_1}{\Delta_i^2}\right)\right\}$. By Lemma C.11 below this implies $\sqrt{C_1\frac{\ln n_i+\ln Kn}{n_i}}\leq\Delta_i/4$ for all $n_i\geq n$, whence $\Delta_i/2-\sqrt{C_1\frac{\ln n_i+\ln Kn}{n_i}}\geq\Delta_i/4>0$. Thus, we can apply the sub-Gaussian assumption to Equation (36) and deduce

$$\sum_{i\in\mathcal{V}^c}\sum_{n_i\geq n}\mathbb{P}\left[\hat{\mu}_{i,n_i}+\sqrt{C_1\frac{\ln n_i+\ln Kn}{n_i}}\geq\mu_i+\Delta_i/2\right]\leq\sum_{i\in\mathcal{V}^c}\sum_{n_i\geq n}\mathbb{P}\left[\hat{\mu}_{i,n_i}-\mu_i\geq\Delta_i/4\right]$$

$$\leq\sum_{i\in\mathcal{V}^c}\sum_{n_i\geq n}e^{-\frac{n_i\Delta_i^2}{32}}$$

$$\leq\sum_{i\in\mathcal{V}^c}\frac{32}{\Delta_i^2}e^{-\frac{n\Delta_i^2}{64}},$$

applying again that $\mathbb{P}\left[\hat{\mu}_{i,n_i}-\mu_i\geq\varepsilon\right]\leq e^{-n_i\varepsilon^2/2}$ for $\varepsilon>0$ and as for Equation (32) we use that $\sum_{n_i\geq n}\exp(-n_ic)\leq\frac{e^{-cn/2}}{c}$ for $c>0$.

**Bounding Equation (34)** First, we can rewrite the conditions $i\in\mathcal{U}(s)$ and $\max_{j\in[K]}n_{0,j}(s)=n$ as $n_{0,i}(s)=n$. Then

$$\sum_{s=n}^{t}\sum_{i\in\mathcal{V}^c}\mathbb{P}\left[i(s)=i, i\in\mathcal{U}(s), \max_{j\in[K]}n_{0,j}(s)=n, n_{0,i}(s+1)=n+1, u_{i*}(s)\leq\mu_{i*}-\Delta_i/2\right]$$

$$\leq\sum_{s=n}^{t}\sum_{i\in\mathcal{V}^c}\mathbb{P}\left[i(s)=i, n_{0,i}(s)=n, \hat{\mu}_i(s)+\sqrt{C_1\frac{\ln n_{i*}(s)+\ln n_0(s)}{n_{i*}(s)}}\leq\mu_{i*}-\Delta_i/2\right]$$

$$\leq\sum_{s=n}^{t}\sum_{i\in\mathcal{V}^c}\mathbb{P}\left[i(s)=i, n_{0,i}(s)=n, \hat{\mu}_i(s)+\sqrt{C_1\frac{\ln n_{i*}(s)+\ln n_{0,i}(s)}{n_{i*}(s)}}\leq\mu_{i*}-\Delta_i/2\right]$$

$$\leq\sum_{s=n}^{t}\sum_{i\in\mathcal{V}^c}\mathbb{P}\left[i(s)=i, n_{0,i}(s)=n, \hat{\mu}_i(t)+\sqrt{C_1\frac{\ln n_{i*}(s)+\ln\frac{n_i(s)}{2}}{n_{i*}(s)}}\leq\mu_{i*}-\Delta_i/2\right] \tag{37}$$

$$+\sum_{s=n}^{t}\sum_{i\in\mathcal{V}^c}\mathbb{P}\left[i(s)=i, n_{0,i}(s)=n, \mathcal{E}_i^c(s)\right], \tag{38}$$

where we use that $n_0(s)\geq n_{0,i}(s)$. In the last step, we consider two cases: either $n_{0,i}(s)\geq n_i(s)/2$ or $n_{0,i}(s)<n_i(s)/2$, which correspond to the conditions $\mathcal{E}_i(s)$ and $\mathcal{E}_i^c(s)$.

We continue with a separate analysis of the two terms in Equations (37) and (38).

**Bounding Equation (37)** We start by reindexing the sum over $s$ in Equation (37) by taking a union bound over all possible values of $n_i$ and $n_{i*}$:

$$\sum_{s=n}^{t} \sum_{i \in \mathcal{V}^c} \mathbb{P}\left[i(s) = i, n_{0,i}(s) = n, \hat{\mu}_i(s) + \sqrt{C_1 \frac{\ln n_{i*}(s) + \ln \frac{n_i(s)}{2}}{n_{i*}(s)}} \le \mu_{i*} - \Delta_i/2\right]$$

$$\le \sum_{i \in \mathcal{V}^c} \sum_{n_i \ge n} \sum_{n_{i*} \ge 1} \mathbb{P}\left[\hat{\mu}_{i,n_i} + \sqrt{C_1 \frac{\ln n_{i*} + \ln \frac{n_i}{2}}{n_{i*}}} \le \mu_{i*} - \Delta_i/2\right]$$

$$= \sum_{i \in \mathcal{V}^c} \sum_{n_i \ge n} \sum_{n_{i*} \ge 1} \mathbb{P}\left[\mu_{i*} - \hat{\mu}_{i,n_i} \ge \sqrt{C_1 \frac{\ln n_{i*} + \ln \frac{n_i}{2}}{n_{i*}}} + \Delta_i/2\right]$$

$$\le \sum_{i \in \mathcal{V}^c} \sum_{n_i \ge n} \sum_{n_{i*} \ge 1} e^{-\frac{n_i}{2}\left(\sqrt{C_1 \frac{\ln n_{i*} + \ln \frac{n_i}{2}}{n_{i*}}} + \Delta_i/2\right)^2}$$

$$\le \sum_{i \in \mathcal{V}^c} \sum_{n_i \ge n} \sum_{n_{i*} \ge 1} e^{-\frac{C_1}{2}\left(\ln n_{i*} + \ln \frac{n_i}{2}\right) - \frac{n_i \Delta_i^2}{8}}$$

$$\le \sum_{i \in \mathcal{V}^c} \sum_{n_i \ge n} \sum_{n_{i*} \ge 1} e^{-\frac{n_i \Delta_i^2}{8}} \left(\frac{1}{n_{i*}}\right)^{C_1/2} \left(\frac{2}{n_i}\right)^{C_1/2}$$

$$= \sum_{i \in \mathcal{V}^c} \left(\sum_{n_i \ge n} \left(\frac{2}{n_i}\right)^{C_1/2}\right) \left(\sum_{n_{i*} \ge 1} e^{-\frac{n_i \Delta_i^2}{8}} \left(\frac{1}{n_{i*}}\right)^{C_1/2}\right)$$

$$\le \sum_{i \in \mathcal{V}^c} \frac{2^{C_1/2+1}}{C_1/2 - 1} \cdot \frac{1}{(n-1)^{C_1/2-1}},$$

where in the fourth line we apply the sub-Gaussian assumption noting that $\sqrt{C_1 \frac{\ln n_{i*} + \ln \frac{n_i}{2}}{n_{i*}}} + \Delta_i/2 > 0$. The next step uses that when $a, b > 0$ then $(a+b)^2 \ge a^2 + b^2$, and that $e^{-x}$ is a decreasing function of $x$. The sixth step follows from noticing that $e^{-\frac{n_i \Delta_i^2}{8}} \ge 1$ as well as $\sum_{n_{i*} \ge 1} e^{-\frac{n_i \Delta_i^2}{8}} \left(\frac{1}{n_{i*}}\right)^{C_1/2} \le \sum_{n_{i*} \ge 1} \left(\frac{1}{n_{i*}}\right)^2 \le 2$ due to $C_1/2 \ge 2$. Further, since $\frac{1}{n_i^c}$ is a decreasing function of $n_i$ we have $\sum_{n_i \ge n} \frac{1}{n_i^c} \le \frac{1}{(c-1)(n-1)^{c-1}}$ for $c > 0$, which is applied choosing $c = C_1/2 - 1 \ge 2$.

**Bounding Equation (38)** Equation (38) can be bounded using the same approach as in Lemma C.2, that is,

$$\sum_{s=n}^{t} \sum_{i \in \mathcal{V}^c} \mathbb{P}\left[i(s) = i, \mathcal{E}_i^c(s)\right] \le \sum_{s=n}^{t} \sum_{i \in \mathcal{V}^c} \mathbb{P}\left[i(s) = i, n_{0,i}(s) < n_i(s)/2\right]$$

$$\le \sum_{i \in \mathcal{V}^c} \sum_{s=n}^{t} \mathbb{P}\left[i(s) = i, n_i(s) - n_{0,i}(s) \ge n_i(s)/2\right]$$

$$\le \sum_{i \in \mathcal{V}^c} \sum_{s=n}^{t} \mathbb{P}\left[i(s) = i, n_i(t) - n_{0,i}(t) \ge n_i(s)/2\right]$$

$$\le \sum_{i \in \mathcal{V}^c} \sum_{n_i \ge n} \mathbb{P}\left[n_i(t) - n_{0,i}(t) \ge n_i/2\right]$$

$$\le \sum_{i \in \mathcal{V}^c} \sum_{n_i \ge n} \frac{2}{(n_i)^{C_2/2-1}} \tag{39}$$

$$\le \sum_{i \in \mathcal{V}^c} \frac{2^{C_2/2}}{(C_2/2 - 2)} \cdot \frac{1}{(n-1)^{C_2/2-2}},$$

where in the third step, we note that $n_i(s) - n_{0,i}(s)$ is a nondecreasing function of $s$. Then, as we only consider the case where $i(s) = i$ and the only term that depends on $s$ is $n_i(s)$, we can reindex by $n_i$. Equation (39) follows from Lemma B.2 and at the end we use similar to above for Equation (37) that $\sum_{n_i \geq n} \frac{1}{n_i^c} \leq \frac{1}{(c-1)(n-1)^{c-1}}$ for $c > 0$ as $\frac{1}{n_i^c}$ is a decreasing function of $n_i$, where we choose $c = C_2/2 - 1 \geq 2$.

**Conclusion**  Combining these results, we finally obtain

$$\mathbb{P}\left[\max_{j \in [K]} n_{0,j}(t) > n\right] \leq \sum_{i \in \mathcal{V}} \frac{8}{\hat{\Delta}_i^2} e^{-\frac{n\hat{\Delta}_i^2}{16}}$$

$$+ \sum_{i \in \mathcal{V}^c} \frac{32}{\Delta_i^2} e^{-\frac{n\Delta_i^2}{64}}$$

$$+ \sum_{i \in \mathcal{V}^c} \frac{2^{C_1/2+1}}{C_1/2 - 1} \cdot \frac{1}{(n-1)^{C_1/2-1}}$$

$$+ \sum_{i \in \mathcal{V}^c} \frac{2^{C_2/2}}{(C_2/2 - 2)} \cdot \frac{1}{(n-1)^{C_2/2-2}}.$$

In particular, for $C_1 = 10$ and $C_2 = 12$ we have

$$\mathbb{P}\left[\max_{j \in [K]} n_{0,j}(t) > n\right] \leq \sum_{i \in \mathcal{V}} \frac{8}{\hat{\Delta}_i^2} e^{-\frac{n\hat{\Delta}_i^2}{16}} + \sum_{i \in \mathcal{V}^c} \left(\frac{32}{\Delta_i^2} e^{-\frac{n\Delta_i^2}{64}} + \frac{32}{(n-1)^4}\right).$$

$\square$

### C.3. General Results on the Selection of $N_i$

In this section, we provide some tools that we use to select $N_i$ in the lemmas needed to derive Theorem 4.1.

**Lemma C.11.** *For constants $C, D, Z \geq 1$, when choosing $N_i = \frac{2CD^2}{\Delta^2} \max\left\{\ln Z, 2\ln\left(\frac{2CD^2}{\Delta^2}\right)\right\}$ it holds that for all $n_i \geq N_i$ that*

$$\sqrt{C \frac{\ln n_i + \ln Z}{n_i}} \leq \frac{\Delta}{D}.$$

To prove this result, we rely on the properties of the Lambert $W$ function and a bound from Chatzigeorgiou (2013, Theorem 1).

**Lemma C.12.** *Let $a, b > 0$ such that $ab \geq e$. Then*

$$\max_x \{x \leq a \ln(bx)\} \leq 2a \ln(ab).$$

*Proof of Lemma C.11.* When $C, D, Z \geq 1$, $\sqrt{C \frac{\ln n_i + \ln Z}{n_i}}$ is a decreasing function of $n_i$. This means that we can focus on finding the smallest value of $N_i$ that satisfies the chosen condition. First note that

$$\min_N \left\{\sqrt{C \frac{\ln N + \ln Z}{N}} \leq \frac{\Delta}{D}\right\} = \min_N \left\{N \geq \frac{CD^2}{\Delta^2}(\ln N + \ln Z)\right\}$$

$$= \max_N \left\{N \leq \frac{CD^2}{\Delta^2}(\ln N + \ln Z)\right\}.$$

Now, we consider two cases:

1. If $N \leq Z$, this implies that $N \leq \frac{CD^2}{\Delta^2}(\ln N + \ln Z) \leq \frac{2CD^2}{\Delta^2}\ln Z$.

2. If $N > Z$ and $N \leq \frac{CD^2}{\Delta^2}(\ln N + \ln Z) \leq \frac{2CD^2}{\Delta^2}\ln N$, then Lemma C.12 gives $\max_N \left\{ N \leq \frac{2CD^2}{\Delta^2}\ln N \right\} \leq \frac{2CD^2}{\Delta^2} 2\ln\left(\frac{2CD^2}{\Delta^2}\right)$.

Combining these results, we deduce that

$$\max_N \left\{ N \leq \frac{CD^2}{\Delta^2}(\ln N + \ln Z) \right\} \leq \frac{2CD^2}{\Delta^2}\max\left\{ \ln Z, 2\ln\left(\frac{2CD^2}{\Delta^2}\right) \right\},$$

which ensures that when picking $N_i \geq \frac{2CD^2}{\Delta^2}\max\left\{ \ln Z, 2\ln\left(\frac{2CD^2}{\Delta^2}\right) \right\}$, then the condition $\sqrt{C\frac{\ln n_i + \ln Z}{n_i}} \leq \frac{\Delta}{D}$ is satisfied for all $n_i \geq N$.

$\square$

*Proof of Lemma C.12.* When $ab > e$, the equation $x = a\ln bx$ has two solutions, which we aim to upper bound. Let $y = bx$, so that we can rewrite our expression as

$$\max_x \{x \leq a\ln(bx)\} = \max_{y/b} \{y/b = a\ln(y)\} = \frac{1}{b}\max_y \{y = ab\ln(y)\}.$$

Then to upper bound the solutions of $y = ab\ln y$ we consider the product log-function (also known as the Lambert $W$ function) $W_{-1}$ and rewrite $y = ab\ln y = -ab\, W_{-1}\left(-\frac{1}{ab}\right)$ with $ab > e$.

By Chatzigeorgiou (2013, Theorem 1), we have that for $ab > e$ it holds that

$$\begin{aligned}
y &= -ab W_{-1}\left(-\frac{1}{ab}\right) \\
&= -ab\, W_{-1}\left(-e^{-\ln\left(\frac{1}{ab}\right)-1}\right) \\
&\leq ab\left(1 + \sqrt{2\ln\left(\frac{ab}{e}\right)} + \ln\left(\frac{ab}{e}\right)\right).
\end{aligned}$$

We deduce that

$$\begin{aligned}
\max_x \{x \leq a\ln(bx)\} &= \frac{1}{b}\max_y \{y = ab\ln(y)\} \\
&\leq a\left(1 + \sqrt{2\ln\left(\frac{ab}{e}\right)} + \ln\left(\frac{ab}{e}\right)\right) \\
&\leq a\left(\sqrt{2\ln\left(\frac{ab}{e}\right)} + \ln(ab)\right) \\
&\leq 2a\ln(ab),
\end{aligned}$$

where the last step follows from comparing the functions $f(x) = \sqrt{2\ln\left(\frac{x}{e}\right)} + \ln(x)$ and $g(x) = 2\ln(x)$ for $x \geq e$. $\square$

## D. Upper Bound in the Non-Realizable and Transitional Cases

In this section, we analyze the behavior of `uncertain-UCB` when the threshold $S$ is either above the expected reward of the best arm, or so close to the threshold that over the course of $T$ rounds these arms continue to contribute to the uncertainty counter $n_0(t)$. For each result in Theorem 4.2, we want to bound the number of times each of the arms contributing to the

regic is played. To do so, for each of these arms $i$, we use a similar decomposition as in Equation (19):

$$\mathbb{E}[n_i(T)] = \sum_{t=1}^{T} \mathbb{P}\left[i(t) = i\right] \leq N_i + \underbrace{\sum_{t=1}^{T} \mathbb{P}\left[i(t) = i, n_i(t) > N_i, u_{i^*}(t) \leq \mu_{i^*}\right]}_{\text{Cases where the best arm } i^* \text{ underperforms}} \tag{40}$$

$$+ \underbrace{\sum_{t=1}^{T} \mathbb{P}\left[i(t) = i, n_i(t) > N_i, u_i(t) \geq \mu_i + \Delta_i\right]}_{\text{Cases where arm } i \text{ overperforms}}, \tag{41}$$

which holds for any $N_i$. We then derive the bounds specified in the following lemmas.

**Lemma D.1.** *When* `uncertain-UCB` *is run under the conditions specified in Theorem 4.2, if arm $i$ is suboptimal and insufficient, then for any $T \geq 1$, Equation (40) is bounded as*

$$\sum_{t=1}^{T} \mathbb{P}\left[i(t) = i, n_i(t) > N_i, u_{i^*}(t) \leq \mu_{i^*}\right] \leq 22.$$

*Furthermore, if arm $i$ is suboptimal and $T \leq \frac{C_1}{4\Delta_*^2}$, then*

$$\sum_{t=1}^{T} \mathbb{P}\left[i(t) = i, n_i(t) > N_i, u_{i^*}(t) \leq \mu_{i^*}\right] \leq 70.$$

*Proof of Lemma D.1.* For any suboptimal arm $i$, we have

$$\sum_{t=1}^{T} \mathbb{P}\left[i(t) = i, n_i(t) > N_i, u_{i^*}(t) \leq \mu_{i^*}\right]$$

$$\leq \sum_{t=1}^{T} \mathbb{P}\left[i(t) = i, n_i(t) > N_i, \hat{\mu}_{i^*}(t) + \sqrt{C_1 \frac{\ln n_{i^*}(t) + \ln n_0(t)}{n_{i^*}(t)}} \leq \mu_{i^*}\right]$$

$$\text{'} \leq \sum_{t=1}^{T} \mathbb{P}\left[i(t) = i, n_i(t) > N_i, \hat{\mu}_{i^*}(t) + \sqrt{C_1 \frac{\ln n_{i^*}(t) + \ln n_{0,i}(t)}{n_{i^*}(t)}} \leq \mu_{i^*}\right]$$

$$\leq \sum_{t=1}^{T} \mathbb{P}\left[i(t) = i, n_i(t) > N_i, \hat{\mu}_{i^*}(t) + \sqrt{C_1 \frac{\ln n_{i^*}(t) + \ln n_i(t)/2}{n_{i^*}(t)}} \leq \mu_{i^*}\right] \tag{42}$$

$$+ \sum_{t=1}^{T} \mathbb{P}\left[i(t) = i, n_i(t) > N_i, n_{0,i}(t) \geq n_i(t)/2\right]. \tag{43}$$

Equation (43) can be bounded by Lemma C.2 as

$$\sum_{t=1}^{T} \mathbb{P}\left[i(t) = i, n_i(t) > N_i, n_{0,i}(t) \geq n_i(t)/2\right] \leq \begin{cases} 16, & \text{if } i \text{ is insufficient,} \\ 64, & \text{if } i \text{ is suboptimal and } T \leq \frac{C_1}{4\Delta_*^2}. \end{cases}$$

For Equation (42), we can reindex the sum over $t$ into two sums over $n_i$ and $n_{i^*}$ and then directly apply the sub-Gaussian

assumption to obtain

$$\sum_{t=1}^{T} \mathbb{P}\left[i(t) = i, n_i(t) > N_i, \hat{\mu}_{i^*}(t) \le \mu_{i^*} - \sqrt{C_1 \frac{\ln n_i(t) + \ln n_i(t)/2}{n_i(t)}}\right]$$

$$\le \sum_{n_i = N_i+1}^{T} \sum_{n_{i^*}=1}^{T} \mathbb{P}\left[\hat{\mu}_{i^*, n_{i^*}} \le \mu_{i^*} - \sqrt{C_1 \frac{\ln n_{i^*} + \ln n_i/2}{n_{i^*}}}\right]$$

$$\le \sum_{n_i = N_i+1}^{T} \sum_{n_{i^*}=1}^{T} \exp\left(-\frac{n_{i^*}}{2} \cdot C_1 \frac{\ln n_{i^*} + \ln n_i/2}{n_{i^*}}\right)$$

$$\le \sum_{n_i = N_i+1}^{T} \sum_{n_{i^*}=1}^{T} \exp\left(-\frac{C_1}{2}\left(\ln n_{i^*} + \ln n_i/2\right)\right)$$

$$\le \sum_{n_i = N_i+1}^{T} \sum_{n_{i^*}=1}^{T} \frac{1}{n_{i^*}^{C_1/2}} \frac{2}{n_i^{C_1/2}}$$

$$\le 6,$$

where the last step follows from $C_1/2 \ge 2$ and $\sum_{t=1}^{\infty} \frac{1}{t^2} \le 2$. Summing our bounds on Equation (42) and Equation (43) finishes the proof. $\square$

**Lemma D.2.** *When algorithm* `uncertain-UCB` *is run under the conditions specified in Theorem 4.2, if arm $i$ is suboptimal and $N_i \ge 8C_1 \frac{\ln T}{\Delta_i^2}$, Equation (41) is bounded by*

$$\sum_{t=1}^{T} \mathbb{P}\left[i(t) = i, n_i(t) > N_i, u_i(t) \ge \mu_i + \Delta_i\right] \le 2.$$

*Proof of Lemma D.2.* For any suboptimal arm $i$, if $N_i \ge 8C_1 \frac{\ln T}{\Delta_i^2}$, we have

$$\sum_{t=1}^{T} \mathbb{P}\left[i(t) = i, n_i(t) > N_i, u_i(t) \ge \mu_i + \Delta_i\right]$$

$$\le \sum_{t=1}^{T} \mathbb{P}\left[i(t) = i, n_i(t) > N_i, \hat{\mu}_i(t) + \sqrt{C_1 \frac{\ln n_i(t) + \ln n_0(t)}{n_i(t)}} \ge \mu_i + \Delta_i\right]$$

$$\le \sum_{t=1}^{T} \mathbb{P}\left[i(t) = i, n_i(t) > N_i, \hat{\mu}_i(t) + \sqrt{2C_1 \frac{\ln t}{n_i(t)}} \ge \mu_i + \Delta_i\right]$$

$$\le \sum_{t=1}^{T} \mathbb{P}\left[i(t) = i, n_i(t) = n_i > N_i, \hat{\mu}_i(t) + \sqrt{2C_1 \frac{\ln t}{n_i}} \ge \mu_i + \Delta_i\right]$$

$$= \sum_{t=1}^{T} \sum_{n_i = N_i+1}^{t} \mathbb{P}\left[i(t) = i, n_i(t) = n_i, \hat{\mu}_i(t) \ge \mu_i + \Delta_i - \sqrt{2C_1 \frac{\ln t}{n_i}}\right].$$

As by assumption $N_i \ge 8C_1 \frac{\ln T}{\Delta_i^2}$, we have $\Delta_i - \sqrt{2C_1 \frac{\ln t}{n_i}} \ge \sqrt{2C_1 \frac{\ln t}{n_i}} > 0$ for all $n_i \ge N_i$, and it follows by sub-

Gaussianity that

$$
\sum_{t=1}^{T} \mathbb{P}\left[i(t) = i, n_i(t) > N_i, u_i(t) \geq \mu_i + \sqrt{2C_1 \frac{\ln t}{n_i}}\right] \leq \sum_{t=1}^{T} \sum_{n_i=N_i+1}^{t} \exp\left(-\frac{n_i}{2} \cdot \frac{2C_1 \ln t}{n_i}\right)
$$

$$
= \sum_{t=1}^{T} \sum_{n_i=N_i+1}^{t} \exp\left(-C_1 \ln t\right)
$$

$$
= \sum_{t=1}^{T} \sum_{n_i=N_i+1}^{t} \frac{1}{t^{C_1}}
$$

$$
\leq \sum_{t=1}^{T} \frac{1}{t^{C_1-1}}
$$

$$
\leq 2,
$$

where the last step uses $C_1 - 1 \geq 2$. $\qquad\square$

With the previous results established, now we are ready to give the proof of Theorem 4.2.

*Proof of Theorem 4.2.* For any suboptimal arm $i$, we pick $N_i = 8C_1 \frac{\ln T}{\Delta_i^2}$. Then we combine Lemma D.1 and Lemma D.2 with the respective definition of regret to prove the different statements of the proof.

**The non-realizable case:** Here, all the suboptimal arms are also insufficient, so for any suboptimal arm $i$, we can apply the first statement of Lemma D.1 together with Lemma D.2 to deduce that

$$
\mathbb{E}[n_i(T)] \leq 8C_1 \frac{\ln T}{\Delta_i^2} + 24.
$$

Plugging this in the definition of the pseudo-regret gives the claimed bound:

$$
R_T = \sum_{i:\Delta_i>0} \Delta_i \, \mathbb{E}[n_i(T)] = \sum_{i:\Delta_i>0} \Delta_i \left(8C_1 \frac{\ln T}{\Delta_i^2} + 24\right) = \sum_{i:\Delta_i>0} \left(8C_1 \frac{\ln T}{\Delta_i} + 24\Delta_i\right).
$$

**The Transitional Regime:** In this setting, some of the suboptimal arms are insufficient but there may also be suboptimal arms that are sufficient. For the first result, we are interested in the satisficing pseudo-regret, and we bound the number of times insufficient arms are played. For any insufficient arm $i$, using Lemmas D.1 and D.2 we obtain as in the non-realizable case before that $E[n_i(T)] \leq 8C_1 \frac{\ln T}{\Delta_i^2} + 24$. It follows that

$$
\mathcal{R}_T^S = \sum_{i:\widetilde{\Delta}_i>0} \widetilde{\Delta}_i \, \mathbb{E}[n_i(T)] = \sum_{i:\widetilde{\Delta}_i>0} \widetilde{\Delta}_i \left(8C_1 \frac{\ln T}{\Delta_i^2} + 24\right) = \sum_{i:\widetilde{\Delta}_i>0} \left(\frac{\widetilde{\Delta}_i}{\Delta_i^2} 8C_1 \ln T + 24\widetilde{\Delta}_i\right).
$$

Finally, we derive refined guarantees for the pseudo-regret when the time horizon $T \leq \frac{C_1}{4\Delta_*^2}$. In this case, we decompose the suboptimal arms into the sufficient and the insufficient ones. For any insufficient arm $i$, we have the same bound as before, i.e., $E[n_i(T)] \leq 8C_1 \frac{\ln T}{\Delta_i^2} + 24$. For a suboptimal arm $i$ that is sufficient, as $T \leq \frac{C_1}{4\Delta_*^2}$, we can apply the second part of Lemma D.1 together with Lemma D.2 to obtain

$$
E[n_i(T)] \leq 8C_1 \frac{\ln T}{\Delta_i^2} + 72.
$$

Plugging these two results into the definition of the regret gives

$$
R_T \leq \sum_{i:\Delta_i>0} 8C_1 \frac{\ln T}{\Delta_i} + \sum_{i:\Delta_i>\Delta_*} 24\Delta_i + \sum_{i:0<\Delta_i\leq\Delta_*} 72\Delta_i,
$$

which finishes the proof.

$\square$

