# OpenReview forum: "Understanding the Gaps in Satisficing Bandits"
_ICML.cc/2026/Conference — ICML 2026 regular_

### Official Review · Reviewer_xA8b · 2026-03-07

**Soundness:** 4
**Presentation:** 4
**Significance:** 3
**Originality:** 3
**Overall Recommendation:** 5
**Confidence:** 4

**Summary:**

This paper studies stochastic satisficing bandits with a known threshold $S$, focusing on how regret depends on three different gap quantities: the standard suboptimality gap $\Delta_i$, the insufficiency gap $S-\mu_i$, and the excess gap $\Delta_\star= \mu_\star-S$. The paper argues that existing guarantees can become misleading when the excess gap is very small, introduces an alternative regret measured against the worst satisfactory arm $\bar \mu$, proves lower bounds for time-independent satisficing algorithms and for the small-excess-gap regime, and proposes uncertain-UCB, which aims to interpolate between bounded satisficing regret in realizable instances and standard UCB-style behavior in transitional and non-realizable regimes.

**Compliance With Llm Reviewing Policy:**

Affirmed.

**Final Justification:**

The clarifications on the initialization issue for $n_0(t)$​ and on the typo/inconsistency in the proof of Theorem 3.3 address my main technical concerns, and I appreciate the additional discussion of the regret metric and of the realizable, transitional, and non-realizable regimes.

I also note the remaining discussion in another review thread regarding the precise framing of Theorem 3.4; from that exchange, it seems that some clarification or reformulation of that theorem may still be needed in the revision.

That said, this does not materially change my overall view of the paper. I still find it interesting, technically grounded, and likely to be useful to researchers thinking about satisficing objectives and the small excess-gap regime. My overall assessment and score therefore remain unchanged.

**Key Questions For Authors:**

1. It would be helpful to provide a stronger justification for treating $\bar R_T^S$ as the primary metric of interest rather than the formally proposed $R_T^S$.

2. A second concern is exposition. The time-independent regime and the transitional regime are central to the paper, but they would benefit from a clearer intuitive explanation, ideally with a simple two-arm example and a sharper sample-complexity interpretation of why $1/\Delta_\star^2$  is the relevant transition scale.

3. Because the claimed advantages are regime-dependent, even a very small synthetic study comparing uncertain-UCB to standard UCB and prior satisficing baselines would substantially strengthen the paper and help the reader see when the proposed guarantees matter in practice.

**Limitations:**

I do not identify any concerns regarding potential societal impact, nor do I see specific societal risks arising from this theoretical work.

**Strengths And Weaknesses:**

Strengths:

Overall, this appears to be a strong paper.

1. The paper targets a genuinely interesting regime that is easy to overlook in prior satisficing work: when $\Delta_\star$ is small, existing guarantees can depend on the “wrong” gap parameter even in instances that otherwise look easy.

2. The overall technical narrative is coherent: the lower bounds motivate why one cannot hope for a uniformly best dependence across all regimes, and the proposed algorithm is designed around exactly that tension.

3. The idea behind uncertain-UCB is appealing. Using an uncertainty counter $n_0(t)$ to behave like UCB early and then effectively stop unnecessary exploration once a sufficient arm is certified is intuitive.

4. The appendix is substantial and suggests serious technical effort.

Soundness

The paper is ambitious and, at a high level, technically interesting. The assumptions are standard for this style of theory paper: sub-Gaussian rewards, a realizable/non-realizable distinction, and a stability condition for one of the lower bounds. The overall methodology also makes sense: the paper first establishes lower bounds, then proposes an algorithm whose design is tied to the structure of those bounds, and finally analyzes the realizable, transitional, and non-realizable regimes separately.
I do have a few clarification requests:

1. Algorithm 1 defines the index using $\ln n_0(t)$, where $n_o(t)$ is the number of "uncertain rounds". As written, it is not obvious that $n_0(t) \geq 1$ always holds after initialization, so the index may be undefined unless some convention is intended.

2. There appears to be at least one inconsistency in Appendix A: in the proof of Theorem 3.3 the means are stated to satisfy $\mu_i = \mu_{i^\star} - \Delta_i >S$ for all $ i \neq i^\star$, but the next sentences discuss “insufficient” arms in that same instance, which seems contradictory unless there is a typo in the inequality sign. These may be fixable presentation issues rather than fatal proof problems, but this should be clarified.

3. A further soundness limitation is that all support is theoretical. For a theory paper this is acceptable in principle, but given how much the paper emphasizes distinct operational regimes, an experimental study would make the claims much easier to validate in practice.

Presentation

The paper is generally well motivated, and the central question is easy to appreciate: prior satisficing guarantees can depend badly on the excess gap even in instances that otherwise seem easy. The introduction does a good job explaining why satisficing matters beyond pure optimization, and Figure 1 is helpful for illustrating how the realizable, transitional, and non-realizable regimes relate as the threshold moves.

I have two concerns regarding the presentation of the paper.

1. The notion of "time-independent" satisficing regret is formally defined, but the paper could do more to explain in plain terms when bounded regret should be possible as opposed to lower/upper bounds which scales with $\log T$ or $\sqrt{T}$.

2. I also think the paper should do more to justify its choice of primary regret metric. The introduction correctly notes that standard satisficing regret is usually threshold-based, i.e. regret is incurred only through shortfall below $S$. The paper then switches attention to the alternative regret $\bar R_T^S$, defined against the worst satisfactory arm, and motivates this choice via a “smooth transition” argument. That is a reasonable analytical perspective, but I was not fully convinced that it is the most natural primary metric for the satisficing objective itself.

Significance

The problem is meaningful. Satisficing objectives are important in settings where meeting a target is more relevant than optimizing to the last decimal, and the paper identifies a real blind spot in previous analyses: small excess-gap instances can make prior guarantees look pessimistic even when the standard bandit gaps are large. I think this is a worthwhile theoretical question, and the paper’s treatment of the transitional regime is a useful contribution because it clarifies how the satisficing problem connects back to ordinary bandit behavior as the threshold approaches the optimal mean.

The impact is likely more specialized than broad, since this is a theory-heavy paper without empirical evidence. Still, for researchers working on bandit theory, satisficing objectives, or regret definitions in structured online learning, the paper could be useful and could influence follow-up work. In particular, the separation between realizable, transitional, and non-realizable regimes feels like a potentially valuable conceptual lens, even if the eventual best regret notion remains open to debate.

Originality

The paper is original in the sense of perspective and analysis rather than in introducing an entirely new algorithmic paradigm. The main novelty lies in isolating the role of the small excess-gap regime, proposing the alternative regret $\bar R_T^S$, proving a refined impossibility result in that regime, and designing uncertain-UCB around an uncertainty counter that adapts across realizable and non-realizable settings.

I would, however, distinguish between what is clearly novel and what is more incremental. The general K-armed lower-bound extension and the transitional-regime analysis seem genuinely useful. The uncertainty-counter mechanism is also a nice twist on UCB. By contrast, the paper’s strongest conceptual departure is the change in regret definition, and whether that counts as a clear advance depends on whether the community finds $\bar R_T^S$ compelling as a primary object rather than just an analytical tool.

---

> ### Author Rebuttal · Authors · 2026-03-31
>
> The authors would like to thank this reviewer for this detailed review. We make sure to take all your comments into account.
>
> The first two points raised in the Soundness section are all valid and will be fixed: the definition of $n_{0}(t)$ will be adjusted to be well defined also for the first rounds; the statement in Theorem 3.3 is a typo and should read $\mu_i = \mu_{i^*} - \Delta_i < S$.
>
> We agree that experimental results will be interesting. The current paper sets out to identify the relevant quantities governing the satisficing regret, enabling meaningful experiments. But it also shows gaps in our understanding. For example, multiple sufficient arms have been ignored in the lower bound. Furthermore, our lower bound with the logarithmic term holds only for $\mathcal{\bar R}^S$, not for $\mathcal{R}^S$. An algorithm more adaptive to the ratio $\Delta_*/\Delta_i$ may be able to bridge these gaps. We believe that a variant of Thompson Sampling that ignores samples below the threshold $S$, might be a candidate, and should be included in experiments.
>
> We appreciate the concerns of most reviewers about our regret metric $\mathcal{\bar R}^S$. Since $\mathcal{\bar R}^S \leq \mathcal{R}^S$, our upper bounds hold for both metrics. One rather technical reason for $\mathcal{\bar R}^S$ is that we were able to show lower bounds for it. A more fundamental reason is that it coincides with the pseudo-regret if there is a single sufficient arm. Thus the threshold can be interpreted as additional information given to the algorithm, leading to time independent pseudo-regret. We will provide a more substantial discussion in the final version.
>
> We will also provide a more detailed explanation of the intuition for time-independence and for the transitional regime.

---

> > ### Author Rebuttal · Reviewer_xA8b · 2026-04-01
> >
> > Thank you for the clear and constructive rebuttal. I appreciate the clarification of the technical issues, as well as the additional discussion regarding the regret metric and the different regimes.
> > The response addresses my main concerns. Overall, my assessment remains unchanged.

---

> > > ### Author Response · Authors · 2026-04-08
> > >
> > > Since in our rebuttal we did not give any details about the intuition for time independent regret and the transitional regime (we ran out of time), I want to elaborate a little.
> > >
> > > In general, rather simple algorithms can achieve time independent regret bounds, if a separating threshold between "good" and "bad" arms is known. One example is the satisficing threshold $S$. For small $\epsilon>0$ consider two arms and two problems with (A) $\mu_1 = S + \epsilon$, $\mu_2=\mu_1-\Delta$ and (B) $\mu_1 =  S - \epsilon$, $\mu_2=\mu_1+\Delta$. Simple bounds can be obtained in terms of $\frac{1}{\epsilon^2}$. To get refined bounds (like our upper bound), more complicated algorithms are necessary, and the intuition below rephrases the idea behind our corresponding lower bound in Theorem 3.4.
> > >
> > > For (B), arm 1 cannot be pulled $1/\epsilon^2$ times (to distinguish it from $\mu_1=S+\epsilon$), as we want to pull the insufficient arm only about $1/\Delta^2$ times. Thus arm 1 can be pulled that often only if it is improbable that arm 2 is actually the sufficient arm.
> > >
> > > To reduce the expected number of pulls of arm 1 from $1/\epsilon^2$ to $1/\Delta^2$, this probability needs to be roughly $\epsilon^2/\Delta^2$. For a bound $P({\hat \mu}_2 < S | (B)) \leq \epsilon^2/\Delta^2$ (the empirical mean ${\hat \mu}_2 < S$ indicating an insufficient arm 2), we need about $\frac{1}{\Delta^2}\log\frac{\Delta}{\epsilon}$ pulls of arm 2. Since a learner cannot reliably distinguish between (A) and (B) before that many pulls of arm 2 (or $1/\epsilon^2$ pulls of arm 1), also for (A) arm 2 needs to be pulled that many times, which gives our lower bound.
> > >
> > > Similar to the $(\log T)$ factor in the standard regret bound, the log factor is not needed to avoid overestimation of bad arms, but to prevent underestimation of good arms: in our case underestimating arm 2 and mistaking (B) for (A) for too many pulls of arm 1.

---

### Official Review · Reviewer_4y8x · 2026-03-12

**Soundness:** 3
**Presentation:** 3
**Significance:** 3
**Originality:** 3
**Overall Recommendation:** 5
**Confidence:** 4

**Summary:**

This paper studies satisficing multi-armed bandits, where the learner aims to play any arm whose expected reward exceeds a known threshold $S$ rather than identifying the optimal arm. The authors focus on the regime where the excess gap $\Delta_*$ (distance from the best arm to the threshold) is small relative to the suboptimality gaps $\Delta_i$, a setting where prior algorithms exhibit regret that scales polynomially in $1/\Delta_*$ even when the problem should be easy from a standard bandit perspective. The paper makes several contributions: (1) a general $K$-armed lower bound of $\Omega(\sum \widetilde{\Delta}\_i / \Delta_i^2)$ extending prior two-armed results; (2) a refined lower bound for the small excess gap regime showing that a logarithmic dependence on $1/\Delta_*$ is unavoidable; (3) a new algorithm, uncertain-UCB, whose satisficing regret scales as $O(\sum (\bar{\Delta}\_i / \Delta_i^2) \ln(K/\Delta_*))$, replacing the polynomial $1/\Delta_*$ dependence of prior methods with a logarithmic one, and featuring a smooth transition to standard UCB1-type guarantees in the non-realizable and transitional regimes (note that this property is also satisfied by other algorithms from the literature). Finally, the authors propose a refined satisficing regret measure using the worst sufficient arm as comparator instead of the threshold.

**Compliance With Llm Reviewing Policy:**

Affirmed.

**Final Justification:**

My main concern regarding the comparison with previous methods have been fully addressed as well as my side questions. I think the overall paper is interesting. While an experimental study would add value to the paper, I don't think it is crucial as long as we can formally compare the algorithms as the authors did in the rebuttal.

**Key Questions For Authors:**

1. Could you provide a formal comparison of prior algorithms' guarantees under your proposed regret measure $\bar{\mathcal{R}}^S_T$?

2. Is the $\ln K$ gap between your upper and lower bounds an artifact of the analysis (the bound $n_0(t) \leq K \cdot \max_j n_{0,j}(t)$) or do you believe it reflects a genuine cost? Could the lower bound be tightened, or could the algorithm's analysis be refined to close this gap?

3. The explicit constants in Theorem C.1 are quite large. Have you considered whether the algorithm is practical at moderate problem sizes, and would you consider adding experiments comparing uncertain-UCB against the algorithms of previous works on instances with varying $\Delta_*$?

**Limitations:**

Yes

**Strengths And Weaknesses:**

Strengths:
- The paper addresses a well-motivated and clearly identified gap in the satisficing bandits literature. The observation that prior algorithms suffer polynomial dependence on $1/\Delta_*$ even on otherwise easy instances is compelling and well-illustrated through the examples in Figure 1.
- The two lower bounds together provide a fairly complete picture of the complexity of satisficing bandits. Showing that no single algorithm can match the one of Theorem 3.3 uniformly is an interesting theoretical contribution.
- The authors provide an algorithm that improves over the dependence on the excess gap in previous methods.

Weaknesses:
- The most significant weakness is the lack of formal comparison with prior methods under unified performance measures. The authors introduce $\bar{\mathcal{R}}^S_T$ but never restate the bounds of Michel et al. (2023), or Feng et al. (2025) under this measure. A comparison table showing $\mathbb{E}[n_i(T)]$ and resulting regret under both $\mathcal{R}^S_T$ and $\bar{\mathcal{R}}^S_T$ for each method would strengthen the paper.
- No empirical evaluation is provided. Experiments would clarify practical relevance, especially given the large explicit constants and makes comparison with prior works clearer.
- The notation is heavy with five gap quantities ($\Delta_i$, $\widetilde{\Delta}\_i$, $\bar{\Delta}\_i$, $\Delta_*$, $\hat{\Delta}\_i$). Definition C.5 uses $\Delta_{i^*}$ which appears to mean $\Delta_*$ (since $\Delta_{i^\ast} = 0$ would make the partition trivial), adding to the confusion.

Issues identified in the proofs:
- In the proof of Theorem 3.3 (Appendix A), the construction states $\mu_i = \mu_{i^\ast} - \Delta_i > S$ for all $i \neq i^\ast$, which would make all arms sufficient. The proof then refers to "insufficient arms," which is contradictory. The intended construction likely requires $\mu_i < S$ for the relevant arms, i.e., $\Delta_i > \Delta_*$.
- In Theorem 3.4, there is a constant mismatch of a factor 8. The proof picks $\alpha = (\sum_j \Delta_*/\Delta_j)^{-1/8}$, yielding $\ln \alpha = \frac{1}{8} \ln(\sum_k \Delta_*/\Delta_k)^{-1}$. The final regret lower bound derived is $\frac{1}{4}\sum_j \frac{\ln \alpha}{\Delta_j} = \frac{1}{32}\sum_j \frac{\ln(\sum_k \Delta_*/\Delta_k)^{-1}}{\Delta_j}$, but the theorem statement claims a factor of $1/4$ instead of $1/32$ in front of the logarithm. The same factor-8 discrepancy appears in the condition on the algorithm's regret. This does not affect the order of the bound, but the precise statement needs correction.
- In the proof of Theorem 4.1, Definition C.5 defines $V = \\{i : \Delta_i \leq \Delta_{i^\ast}/2\\}$, but $\Delta_{i^\ast} = 0$, making $V = \emptyset$. This is probably meant to be $\Delta_*$. Furthermore, the claim that $N_{0\max} = 64/\Delta_*^2$ satisfies $N_{0\max} \geq \max_{j \in V^c} 64/\Delta_j^2$ requires $\Delta_j \geq \Delta_*$ for all $j \in V^c$, but sufficient arms in $V^c$ can have $\Delta_*/2 < \Delta_j < \Delta_*$. This is fixable by increasing $N_{0\max}$ by a constant factor, preserving the order of the bound but requiring updated constants.
I believe that none of these issues invalidate the main contribution.

---

> ### Author Rebuttal · Authors · 2026-03-31
>
> The authors would like to thank this reviewer for this detailed review and for identifying all of these small errors, which you have correctly identified as being typos. We will make sure to correct them in the final version.
>
> You are right about the lack of formal comparison with the previous methods.
>     A mixed bound as suggested by the second reviewer shows that our bound improves on the previous bounds, and in some cases quite substantially,
>
> $\mathcal{\bar{R}}^S = O(\sum_i \frac{1}{\Delta_i} \log(\frac{K}{\Delta_*})) =O(\sum_{i:\tilde \Delta_i \leq \Delta_*}
> \frac{1}{\Delta_*} \log(\frac{K}{\Delta_*})     + \sum_{i:\tilde \Delta_i > \Delta_*}  \frac{1}{\tilde \Delta_i}
> \log(\frac{K}{\Delta_*})).$
>
> Since $\mathcal R^S \leq \mathcal{ \bar R}^S$ this also bounds $\mathcal{R}^S$. For $\tilde \Delta_i \ll \Delta_*$, this improves significantly over Michel et al. For $\tilde \Delta_i \gg \Delta_*$, this improves significantly over both Michel et al. and Feng et al. For $\tilde \Delta_i \sim \Delta_*$, we have an additional log factor that does not appear in Michel et al.
>
> Concerning the lower bound: We believe that the factor $K$ is an artifact of the analysis or rather the algorithm. In the analysis, this $K$ factor reflects the effect of having potentially many arms close to the threshold, that would initially increase the uncertainty whenever they are played.
> It seems possible to replace the count of the uncertain rounds $n_0$ in the algorithm by the max of the uncertain rounds when arm $i$ was played, $\max_i n_{0,i}$, without affecting the proof.
>
> Concerning experiments and practical application: We have not conducted experiments, yet. But this is certainly of interest for future work, in particular in comparison with maybe more practical algorithms, like a variant of Thompson Sampling which discards samples below threshold $S$.

---

> > ### Author Rebuttal · Reviewer_4y8x · 2026-04-01
> >
> > Thank you for your answer. My main concern regarding the comparison with previous methods have been fully addressed as well as my side questions. I think the overall paper is interesting. While an experimental study would add value to the paper, I don't think it is crucial as long as we can formally compare the algorithms as the authors did in the rebuttal. I will increase the score accordingly.

---

### Official Review · Reviewer_V7f1 · 2026-03-14

**Soundness:** 2
**Presentation:** 2
**Significance:** 3
**Originality:** 3
**Overall Recommendation:** 3
**Confidence:** 4

**Summary:**

$\newcommand{\td}{\tilde{\Delta}} \newcommand{\bd}{\bar\Delta} \newcommand{\brs}{\bar{\mathcal{R}}^S} \newcommand{\rs}{\mathcal{R}^S} \newcommand{\r}{\mathcal{R}}$

This paper is on low-regret satisficing multi-arm bandit problems. Let $\mu\_i$ denote the means in a bandit instance as usual, and let $S$ be a given threshold. An arm is called sufficient if $\mu\_i \ge S,$ and insufficient otherwise. The goal in this problem is to limit a satisficing notion of regret which does not penalize plays of sufficient arms.

In more detail, define the following three notions of gaps:
- $\Delta\_i = \mu\_* - \mu_i$ is the usual gap, where $\mu\_* = \max\_i \mu\_i$.
- $\td\_i = S - \mu\_i$ is the gap to the threshold.
- $\Delta_* = \mu\_* - S$ is the slack at the optimum. Note that $\Delta\_* = \td\_i + \Delta\_*$.

The goal is to control $\rs\_T = \sum_{i : \td\_i > 0} \td\_i \mathbb{E}[n\_i(T)]$ (when $\Delta\_* > 0$). Ideally, the method to do so should have two important properties:
- When $\Delta\_* > 0, \brs\_T$ should not diverge as $T \to \infty$, i.e., there should be some constant bound (in terms of the $\Delta$s) on $\brs\_T$. This is the "realizable" case.
- When $\Delta\_* < 0,$ the usual regret $\r\_T = \sum_{i : \Delta\_i > 0} \Delta\_i \mathbb{E}[n\_i(T)]$ should be well-controlled. This is the "non-realizable" case.

This problem has seen some interest in the recent literature, with two pertinent results being those of Michel et al., who provide a method with $\rs\_T = O( \sum 1/\td\_i),$ and of Feng et al., who provide a method with $\rs\_T = O( K/\Delta\_* \textrm{polylog}(K/\Delta\_*))$ and further show UCB-scale regret bounds for non-realizable cases.

This paper constructs both minimax gap-dependent lower bounds for this problem, and designs a simple variation of UCB that attains nearly tight gap-dependent upper bounds on $\rs$, as well as UCB-scale bounds on regret in non-realizable problems.

The lower bounds are all based on instances wherein there is only one sufficient arm. It is first argued that (under a mild assumption on the algorithms), insufficient arms must be pulled at least $\Delta\_i^{-2}$ times each, which directly gives minimax bounds. A further refinement is offered in the regime where $\Delta\_\*$ is very small, in which case essentially an additional $\log(1/\Delta\_\*)$ dependence is found.

The upper bound is based on a UCB method, where the bonus in the confidence bound is driven by the number of rounds in which an arm that is potentially insufficient (in the sense that the LCB is below $S$) is played, instead of with time. The key effect of this is that in the large $t$ limit, as long as a sufficient arm is identified, this $n\_0(t)$ would saturate, and thus prevent the UCBs of insufficient arms growing with $t$. On the other hand, if the instance is non-realizable, $n\_0(t) = t$, and a standard UCB behaviour is recovered. Much of the challenge in the analysis is in tracking the behaviour of $n\_0(t)$ through the run, for which certain augmentations to the standard UCB analysis are constructed. The net result is that any insufficient arm $i$ is played at most $O(\Delta\_i^{-2} \log(K/\Delta\_*))$ times, where the $\log$ term arises essentially due to the contribution of sufficient arms to $n_0(t)$.

**Compliance With Llm Reviewing Policy:**

Affirmed.

**Key Questions For Authors:**

Key question regarding evaluation hinge on the role of $\bar{\mathcal{R}}^S,$ and the issues with Thm 3.4 as outlined above.

I have a non-key question as well:

- It seems like the basic idea of replacing the "clock," $t$, in standard bandit approaches by the $n_0(t)$ should also work for other UCB style ideas (like KL-UCB). Have you explored such generalizations?

**Limitations:**

Yes

**Strengths And Weaknesses:**

I think this is a nice paper. The problem is natural and interesting, and has seen attention in the recent literature. I found the uncertain-UCB method especially elegant, both since it is a small modification of the UCB approach (and thus may lend itself to generalizations to richer settings), and because it recovers the bounds of both Michel et al. and Feng et al. in appropriate regimes (up to the $\log(1/\Delta\_\*)$ factor) as well as the natural regret guarantee in the unrealizable setting. The minimax bound of section 3.3 is also interesting, although the construction leaves one wondering what exactly is tight for problems with many sufficient arms. I read through the proofs, and the results of Thms 3.3, 4.1, and 4.2 are correct, to my understanding. Most of the technical exposition is clear.

There are two main grouses I have with the paper.

- The first is to do with the metric $\brs$, whose point I just don't understand. It appears to neither be inherently natural for the satisficing bandit problem (certainly this case is not made), and it also appears to be not particularly important for the argument of the results. I think introducing this distracts from the results of the paper.

    In the context of the order-wise theory being developed in the paper, the only time that $\brs$ is more than a constant factor off from $\rs$ is when $\Delta\_\*$ is very big compared to the $\td_i$s. However, none of the results actually explore this regime in a way that is critical to them. Thms. 3.3, 4.1, 4.2 all directly control the play of insufficient arms (from below or above as appropriate), and so can directly be used to give bounds on $\brs$ or $\rs$ (in particular, Thms. 4.1, 4.2 can be stated for $\rs$ with only cosmetic changes). Thm. 3.4 is explicitly in the regime $\frac{1}{\Delta\_\*} > 2\sum \frac{1}{\Delta\_i},$ which in turn implies that $\Delta\_\* \le \Delta\_i/2$ for all $i$. But since $\Delta\_i = \td\_i + \Delta\_\*,$ this immediately implies that $\bd\_i \le 2\td\_i,$ so this result can be converted into one for $\rs$ as well.

    As such the only role of $\rs$ is that the argument of Thm. 3.4 finds its use more convenient, but this hardly a reason to introduce a new metric into the literature that deviates from the natural goal of playing arms with mean above $S$. In fact, since the construction of Thm. 3.4 uses constellations with only one sufficient arm, in this proof, $\brs$ can be replaced by just the usual regret (along with the observation that in this setting, regret is at most twice $\rs$).

- The second is to do with Thm 3.4, which is stated in a misleading way, and does not quite capture an appropriate minimax bound.

    The issue is a little subtle. In the instance $\nu'$ constructed in the proof, the gaps are different than the in the instance $\nu$. Indeed, for any $j \neq i,$ if I denote the gaps under $\nu'$ as $\Delta'$ we have $ \Delta\_j' = \Delta\_j + \Delta\_i - 2\Delta\_\*$, and $ \Delta\_\*' = \Delta\_i - \Delta\_\*.$

     This makes the statement kind of strange: the statement fixes a set of gaps and a particular arm instance with those gaps. The argument says that if $\brs$ for this instance is smaller than some function of the gaps, then there exists an instance (with different gaps) for which the regret is bigger than this function of the original gaps (up to a constant). I don't know why this is a minimax bound in terms of gaps.

    This would be more convincing if it were instead shown that $\brs\_T(\nu') \ge f(\Delta'),$ where $f(\Delta)$ is the expression in the statement. It is unclear if this would work. Of course, $\Delta\_j' > \Delta\_j$ for each $j,$ which is good, but the issue is that $\Delta\_\*' > \Delta\_\*$ as well. Then it is not even obvious that, e.g., $\sum \Delta\_\*'/\Delta\_j' < 1$ (and this can fail: if all the $\Delta\_i$s are identical, then this sum goes to $\ge (K-1)/2$ as $\Delta\_\* \to 0$). Maybe this can be salvaged, I don't know.

    However, even if this all worked with $f(\Delta')$, it is unclear what is being shown. Globally, this works out to something like "there is a sequence of gaps $\{\Delta\_i\}$ and a $\Delta\_\*$ such that any method must have $\brs\_T \ge f(\Delta)$. Certainly this is not nearly as local a result as is suggested by the statement of Thm 3.4, and is also weaker in flavour than a standard bandit bound, which is more of the form "for any gap sequence, any algorithm must have regret $g(\Delta) \log(T)(1-o(t))$.


Overall, I think that the paper needs some significant rewrites: both to address the issues of Thm 3.4, and also to either eliminate $\brs,$ or to justify its use more strongly. These are kind of major changes: they affect the main claimed contributions, and addressing these would require rewriting nontrivial parts of the paper. For this reason, I think this paper is not ready for publication yet, which drives my recommendation below. I do want to say that with some more work this paper would be a nice contribution.


----

Some minor points:

- It would be helpful to add a discussion of the bound of Thm. 4.1 after the statement. In particular, by observing that $\Delta\_i = \td\_i + \Delta\_\*$, at least if you state it in terms of $\rs,$ you can observe that it recovers the guarantees of Michel et al. up to the log-factor, and also those of Feng et al. when the $\td\_i$s are smaller than $\Delta\_\*$. It might also be useful to explicitly state the bound on the expected number of plays of insufficient arms, because then you can write down, as corollaries, mixed bounds like $$\frac{\rs}{\log(K/\Delta\_\*)} \le C \sum\_{i : \td\_i > \Delta\_\*} \frac{1}{\td\_i} + C\sum\_{i : 0 < \td_i < \Delta\_\*} 1/\Delta\_\*.$$ Such discussions should make the strength of the result relative to the literature much clearer.

- In the proof of Thm 3.3, I think line 510 should say $\mu_i = \mu_{\*} - \Delta_i < S$ instead of $>S$.

---

> ### Author Rebuttal · Authors · 2026-03-31
>
> We thank the reviewer for the thoughtful comments.
>
>  We fully agree with the reviewer that all upper bound results could have been stated in terms of $\mathcal{R}^S$ instead of $\mathcal{\bar R}^S$, which we will make clear in the final version of the paper. To us, $\mathcal{\bar R}^S$ felt a bit more natural, as for a single sufficient arm it equals the pseudo-regret. We believe that this is a matter of opinion. For the lower bound in Theorem 3.4,  $\mathcal{\bar R}^S$ is required.
>
>
> Theorem 3.4 is indeed not a typical minimax lower bound. Essentially, it states that, for any algorithm, there are settings for which the algorithm performs worse than the lower bound.
>  More formally, the reason for this less strict lower bound is that the problem becomes easier for known gaps. For example, if a lower bound $\tilde{\Delta} \leq \tilde{\Delta}_i$ is known, then $\mathcal{R}^S$ and $\mathcal{\bar R}^S$ can be bounded by $K/\tilde{\Delta}$ without the log factors, similar to Theorem 1 in Bubeck et al (2013). Thus, the unusual type of lower bound is necessary, since only not knowing the magnitude of the gaps incurs the log factor.
>
>
> Note that for our lower bound, we assume a bandit problem with small $\Delta_* \ll \Delta_i \sim \tilde\Delta_i$.
> We then construct a modified bandit problem with $\tilde\Delta_i' \sim \Delta_*$ and $\Delta_i' \sim \Delta_i$, such that $\tilde\Delta_i' \ll \Delta_*' \sim \Delta_i'$. We derive a lower bound on $\mathcal{\bar R}^S$, but since $\tilde{\Delta}_i' \ll \Delta_i'$, this does not translate into an interesting bound on $\mathcal{R}^S$. It is an open question whether a similar lower bound for $ \mathcal{R}^S$ can be derived.
>
> Thank you for the suggestion of stating mixed bounds.
>
>  We haven't explored the generalizations of the "uncertainty clock" to other algorithms yet, this is certainly a promising direction for future work.

---

> > ### Author Rebuttal · Reviewer_V7f1 · 2026-04-02
> >
> > The rebuttal addresses none of my criticisms in a substantial way. $\newcommand{\brs}{\overline{\mathcal{R}}^S}$ Regarding Thm. 3.4, the rebuttal ignores the issue that a different gap structure in $\nu'$ means that a result in terms of the gaps in $\nu'$ is required in order to assert that a logarithmic dependence on the gap of the top arm from the threshold is needed, even in the weak sense outlined above. Of course, the weakness of the sense in which the lower bound is stated is not contested. Further, regarding $\mathcal{R}^S$ versus $\brs$, the reason given is that the statement of Thm. 3.4 is interesting for the latter, but not the former. This is suspect for two reasons---firstly, that Thm 3.4 is not watertight itself, and secondly that it is trivial to derive the same bound for $\mathcal{R}^S$ using the results in the paper, as I outline below.
> >
> > Overall, then, my opinion remains the same---the statement of Thm 3.4 has conceptual issues, and the introduction of $\brs$ is poorly justified. $\newcommand{\bd}{\mathbf{\Delta}}$
> >
> > ---
> >
> > Regarding Theorem 3.4.
> >
> > I raised two issues here: the main one was that the lower bound for the $\nu'$ instance was not in terms of hte gaps $\Delta'$, but in terms of the gaps for a different instance, making the bounds not meaningful. The less important one was that the bounds were quite weak, in only showing existence of gap sequences where $\brs$ showed a $\log(1/\Delta\_*)$ dependence. The second issue was not contested.
> >
> > For the main issue, the rebuttal ignores it. As far as I understand it, the substantial claim desired is that a worst-case $\log(1/\Delta\_\*)$ dependence is necessary. For this, an instance $\nu$ with a certain gap sequence and $\Delta\_*$ is taken. I will write these parameters together as $\bd$. Then a different instance $\nu'$, with a difference gap sequence, which I will write as $\bd',$ is constructed. Then the claim is that $\brs(\nu) < f(\bd) \implies \brs(\nu') \ge f(\bd)/4$ for a function $f$. My criticism is that there is a $f(\bd)$ sitting in the bound on $\brs(\nu')$ there instead of $f(\bd')$. In particular, the convincing claim here would be that $\brs(\nu') \ge \sum (\Delta'\_i)^{-1} \Omega( \log 1/\Delta\_\*')$---only then can one even assert that there are instances for which a logarithmic dependence on the gap of the optimal arm from the threshold is necessary.
> >
> >    Let me illustrate the point with an example. Suppose we are interested in the standard bandit problem, and we want to show that in general the regret must scale with the sum of the inverse gaps. To this end, I take an instance $\nu$ with a certain value of $\sum \Delta\_i^{-1}.$ Then I construct a different instance $\nu'$, where $(\Delta'\_2)^{-1} >  \sum \Delta\_i^{-1}$. Finally, I claim that in the instance $\nu', $ regret is more than $\log(T) (\Delta'\_2)^{-1} > \log(T) \sum \Delta\_i^{-1}.$ This is not telling me that a worst-case dependence on the sum of inverse gaps is necessary---the gap structure in the two instances is so different that even the weaker bound $\log(T) (\Delta'\_2)^{-1}$ for $\nu'$ is larger than the sum of inverse gaps in $\nu$, but that does not mean that in $\nu'$ we have a dependence on the sum of inverse gaps of $\nu'$.
> >
> > Given this, even in the weak sense demanded, the theorem does not say anything meaningful.
> >
> > ---
> >
> > Regarding $\brs$ versus $\mathcal{R}^S$:
> >
> > First, I don't think it is just a matter of opinion which one to use. The metric associated with a problem is an important component of its standardisation. Changing it is not a casual matter---it adds confusion to the literature, and obscures connections. It is fine to propose a new metric if it brings new insights, but that is exactly what I was questioning here.
> >
> > In the rest of the rebuttal, the point made is that while Thm. 3.4 (if we accept it at face value, which I do not) would be an interesting result in terms of $\brs$, it would not be interesting in terms of $\mathcal{R}^S$. I don't see this either---in fact the same result holds for $\mathcal{R}^S$, up to constants. See below.
> >
> > By the hypothesis, that $\Delta\_\* < \min\_i \Delta\_i/2$. Thus $\brs(\nu) \le 2\mathcal{R}^S(\nu)$ (see the review). Take the hypothesis that $\mathcal{R}^S < 1/2 \cdot R,$ where $R$ is the right hand side of the condition on line 232 col 2. This enables the entirety of the proof started on page 5. Following this argument verbatim, we get to $\mathbb{E}\_{\nu'}[ N\_1(T)] \ge \log(\alpha)/( 4 \Delta\_\*^2 \alpha^8)$ (line 295 col 2). Now, since the threshold $S$ is the same, and since $\mu\_1' = S- \Delta\_\*,$ you also have $\tilde{\Delta}\_1' = \Delta\_\*$. Immediately, we get $\mathcal{R}^S(\nu') \ge \tilde{\Delta}\_1' \mathbb{E}\_{\nu'}[N\_1(T)] \ge \log(\alpha)/(4 \Delta\_* \alpha^8),$ which is exactly the bound stated for $\brs$. Now one can finish identically.

---

> > > ### Author Response · Authors · 2026-04-05
> > >
> > > We are very grateful for your comments, which have helped us to clarify and even improve our results. Indeed, the current statement of Theorem 3.4 is vacuous, as it does not relate its regret bound to the modified problem setting $\nu'$. Furthermore, your suggestion for adapting the proof indeed allows to derive the same bound also for $\mathcal{R}^S(\nu')$.
> > >
> > > To relate the bound of Theorem 3.4 to the parameters of $\nu'$, we need to use the construction of $\nu'$ in the proof. Let us consider the case of only two arms, with the sufficient arm $\mu_1=S+\Delta_*$ and the insufficient arm $\mu_2=\mu_*-\Delta_2 <S$. Then $\nu'$ is constructed as $\mu_1' = S - \Delta_*$ and $\mu_2' = \mu_1'+\Delta_2$, with $\Delta_1'=\Delta_2$,
> > > $\tilde\Delta_1'=\Delta_*$,
> > > and $\Delta_*'=\Delta_2-\Delta_*$.
> > >  The bound of the theorem gives
> > >
> > > $$
> > > \mathcal{\bar R}^S(\nu')
> > > \geq\frac{\ln \frac{\Delta_2}{\Delta_*}}{4\Delta_2}
> > > = \frac{\ln \frac{\Delta_1'}{\tilde\Delta_1'}}{4\Delta_1'}
> > > \geq \frac{\ln \frac{e\Delta_1'}{\Delta_*'}}{4\Delta_1'}
> > > $$
> > >
> > > for $\Delta_*$ sufficiently small.
> > > This allows to reframe the theorem (for two arms): For any algorithm, any $\Delta$ and any small $\gamma$,
> > > \begin{align*}
> > > \mathcal{\bar R}^S(\nu')
> > > \geq \ln(1/\gamma)/(4\Delta)
> > > \end{align*}
> > > for $\mu_1=S+\gamma\Delta$, $\mu_2=\mu_1-\Delta$, $\mu_*=\gamma\Delta$, or for $\mu'_2=S+(1-\gamma)\Delta$, $\mu'_1=\mu'_2-\Delta$, $\tilde{\Delta}_1'=\gamma\Delta$.
> > >
> > > This has the following interpretation: If the sufficient arm is close to $S$, then there is little statistical evidence against it being insufficient and close to $S$, instead.  Thus, the other arm (which could be sufficient) needs to be protected against underestimation by using this extra logarithmic factor for the number of samples.
> > >
> > >
> > > For general $K$ the magnitudes of the $\Delta_i$ need to be taken into account, but an immediate bound can be obtained for equal $\Delta_j=\Delta$, $j=2,\ldots, K$:
> > >
> > > \begin{align*}
> > > \mathcal{\bar R}^S(\nu')
> > > \geq (K-1)\frac{\ln\frac{1}{(K-1)\gamma}}{4\Delta}
> > > \end{align*}
> > > for $\mu_1=S+\gamma\Delta$, $\mu_j=\mu_1-\Delta$, or for some $\mu'_i=S+(1-\gamma)\Delta$, $\mu'_1=\mu'_i-\Delta$, $\mu'_j=\mu_j=\mu'_i-2(1-\gamma)\Delta$, $j \neq i$.

---

### Official Review · Reviewer_txqf · 2026-03-16

**Soundness:** 4
**Presentation:** 3
**Significance:** 3
**Originality:** 4
**Overall Recommendation:** 5
**Confidence:** 3

**Summary:**

The paper studies a variant of the standard multi-armed bandit problem called satisficing bandits, in which the goal of the learner is to identify and play actions that have mean reward above a given threshold value, as opposed to actions with maximal mean reward. A different notion of the satisficing pseudo-regret is introduced, which replaces the gaps between mean rewards of insufficient arms and the threshold by the gaps between the mean rewards of the insufficient arms and the mean reward of the worst sufficient arm. The first result is a general asymptotic lower bound for realisable satisficing K-armed bandits. This is complemented by a refined lower bound that is specialised to the the small excess gap regime, and which shows that no algorithm can match the general lower bound across all instances. In particular, there are instances with a small excess gap for which the general lower bound is unattainable. An algorithm called uncertain-UCB is proposed, which is designed to perform well in three cases: realisable problems with several sufficient arms; realisable problems with a single sufficient arm (the transition regime); non-realisable problems. It is proved that uncertain-UCB simulaneously has: almost optimal satisficing pseudo-regret in any realisable problem; regret and satisficing pseudo-regret of order log(T) in the transition regime with a small excess gap; regret of order log(T) in non-realisable problems.

**Compliance With Llm Reviewing Policy:**

Affirmed.

**Final Justification:**

My initial assessment of the paper was positive. I thought that the paper was well-written and that the main results of the paper were interesting. I raised a couple of clarifying questions, and these were resolved by the authors' rebuttal. In my initial review, I hadn't noticed that the lower bound in Theorem 3.4 was stated entirely in terms of the gaps of a single instance $\nu$, which, as pointed out by Reviewer V7f1, made the claim of Theorem 3.4 as originally stated vacuous. However, I have read the authors' response to this flaw and as far as I can tell, it resolves the issue. Overall, my impression of the paper remains positive, so I am maintaining my positive score.

**Key Questions For Authors:**

In the paragraph following Theorem 3.3, it is stated that this lower bound shows that the upper bound in Michel et al. (2023) is optimal up to constants when the ratio between tilde(Delta)i and Delta* is constant. How does a constant ratio between tilde(Delta)i and Delta* make 1/Delta_i appear in the upper bound in Michel et al. (2023)?

Is there any hope of improving the constant factor in the non-realisable setting such that the regret of (a version of) uncertain-UCB has the asymptotically optimal constant factor in front of the log(T) term?

**Limitations:**

Yes.

**Strengths And Weaknesses:**

*Strengths:*
I found the lower bounds (and especially theorem 3.4) to be genuinely interesting results that provide new insights about the problem.

The paper is very well-written. The introduction did a good job of introducing the problem and describing the limitation of previous results. Despite being quite technical, the proof of Theorem 3.4 was easy to follow. The proof sketch for the upper bound did rather a good job of compressing the full proof. The verbal description of the uncertain-UCB algorithm was effective in conveying the motivation behind the design of the algorithm.

*Weaknesses:*
I don’t think that the paper has any major weaknesses.

Something that bothered me a little is that the number of times arm i is played in T rounds is sometimes denoted by a lower case n and sometimes by an upper case N.

I found a few possible typos. On line 120, column 1, I think “Algorithm and analysis…” should be “The algorithm and analysis…”.  In the statement of Theorem 3.3, the regret has a subscript of lower case t, but the limit is taken w.r.t. upper case T. On line 273, column 2, I think there is missing capitalisation in “equation (6)”. In the definition of the event Ei(t), “n_{0,}” should be “n_{0,i}”.

A non-weakness is that I don’t like the word “satisficing”, but this paper can’t be blamed for that.

*Overall:* I like this paper.

---

> ### Author Rebuttal · Authors · 2026-03-31
>
> The authors thank the reviewer for the encouraging review and for identifying typos, which we will fix in the final version of the paper.
>
> Regarding the first question, the sentence ''constant ratio" was indeed unclear and should be replaced by ''$\tilde \Delta_i$  and $\Delta_*$ have the same order of magnitude'', $\widetilde \Delta_i \sim \Delta_*$. Then $\Delta_i = \widetilde \Delta_i + \Delta_* \sim \tilde \Delta_i \sim \Delta_*$ and our lower bound and the upper bound of Michel et al. coincide.
>
> In contrast, if $\tilde \Delta_i \sim \Delta_*$ implying $\Delta_i \sim \Delta_*$, or $\tilde \Delta_i \gg \Delta_*$ implying $\Delta_i \sim \tilde \Delta_i$, then  the upper bound of Michel et al. is much larger than our lower bound, $\sum_i \frac{1}{\widetilde \Delta_i} + \frac{\widetilde \Delta_i}{\Delta_*^2} \gg \sum_{i} \frac{\widetilde \Delta_i}{\Delta_i^2}$.
>
> Regarding the second question,  the large constants are currently an artifact of the analysis.
>
> It might be possible to recover the optimal constants in the unrealizable case by upper bounding the log term in the UCB bound of our algorithm, $C_1(\ln n_i(t) + \ln n_0(t))$, by an optimal UCB term for the standard problem.

---

> > ### Author Rebuttal · Reviewer_txqf · 2026-04-04
> >
> > Thank you for your answers to my questions. The points raised in my own review have been fully resolved.
> >
> > Regarding Theorem 3.4, I hadn't noticed that the lower bound is of the form $\mathcal{R}(\nu) < f(\Delta) \implies \mathcal{R}(\nu^{\prime}) \geq f(\Delta)/4$ as opposed to $\mathcal{R}(\nu) < f(\Delta) \implies \mathcal{R}(\nu^{\prime}) \geq f(\Delta^{\prime})/4$. If this could be fixed, then I still think that the statement of Theorem 3.4 would be meaningful. Otherwise, I agree with Reviewer V7f1 that this lower bound is not so useful after all.

---

> > > ### Author Response · Authors · 2026-04-05
> > >
> > > Thank you for your concern. We have addressed the question about Theorem 3.4 in our response to reviewer V7f1. While the statement of the theorem fails to connect the derived lower bound to the relevant quantities of $\nu'$, the construction of $\nu'$ in the proof gives ${\cal R}(\nu') \geq f(\Delta,\Delta_*)/4 \geq f(\Delta',\Delta_*')/4$.

---

### Decision · Program_Chairs · 2026-04-30

**Decision:**

Accept (regular)

**Comment:**

This paper considers satisficing in bandit problems where, instead of aiming to minimize the regret with respect to the optimal arm, the learner is content with meeting a target level of performance.

During the review and discussion phases, the reviewers carefully examined the proofs, identified issues, and communicated them effectively with the authors. The authors also responded well to the concerns of the reviewers and clearly explained the points of confusion and provided procedures to fix the issues.

After the rebuttal and discussion, it is concluded that this paper makes a significant contribution to the growing literature on satisficing bandits by proposing (i) an interesting regime where previous algorithms can be ineffective, (ii) an intuitive but minor modification of UCB that works well, (iii) a lower bound (although somewhat weak) for the considered problem.

I recommend acceptance. The authors should clearly address all concerns raised by the reviewers (as they did in the rebuttal) in the final version of the paper. In particular, please pay attention to the following points.

I find the concern about the representation of results in terms of a rather unnatural regret term $\bar{{\cal R}}^S$ instead of the natural and usual regret term ${\cal R}^S$ important, as it obscures the presentation of the results and their straightforward comparison with the literature. This issue can be easily fixed by rewriting parts of the paper to center on ${\cal R}^S$ without changing its technical content.

The concerns regarding Theorem 3.4 are also very important, as it is presented as one of the key theorems in the paper. The way the authors fixed Theorem 3.4 appears technically correct, but it seems that the current fix does not close the gap between the lower bound and the upper bound for their algorithm. Despite this weakness, I think that the contribution of this paper is significant enough. However, this issue should be carefully highlighted, and the limitations of the lower bound should be clearly discussed in the final version.